# The transcription factor HBP1 promotes ferroptosis in tumor cells by regulating the UHRF1-CDO1 axis

**Ruixiang Yang[1], Yue Zhou[1], Tongjia Zhang[1], Shujie Wang[1], Jiyin Wang[1], Yuning Cheng[1], Hui Li[1], Wei Jiang[1], Zhe Yang[2], Xiaowei Zhang****[1]\***

**1** Department of Biochemistry and Biophysics, School of Basic Medical Sciences, Beijing Key Laboratory of Protein Posttranslational Modifications and Cell Function, Peking University Health Science Center, Beijing, China, **2** Department of pathology, The First Affiliated Hospital of Kunming Medical University, Kunming, Yunnan Province, China

\* xiaoweizhang@bjmu.edu.cn

**Data Availability Statement:** The analysis code and datasets supporting the conclusions of this article are available in additional supplementary file S1 Data. All flow cytometry data for Figs 4E, 7F,

## Abstract

The induction of ferroptosis in tumor cells is one of the most important mechanisms by which tumor progression can be inhibited; however, the specific regulatory mechanism underlying ferroptosis remains unclear. In this study, we found that transcription factor HBP1 has a novel function of reducing the antioxidant capacity of tumor cells. We investigated the important role of HBP1 in ferroptosis. HBP1 down-regulates the protein levels of UHRF1 by inhibiting the expression of the *UHRF1* gene at the transcriptional level. Reduced levels of UHRF1 have been shown to regulate the ferroptosis-related gene *CDO1* by epigenetic mechanisms, thus up-regulating the level of CDO1 and increasing the sensitivity of hepatocellular carcinoma and cervical cancer cells to ferroptosis. On this basis, we constructed metal-polyphenol-network coated HBP1 nanoparticles by combining biological and nanotechnological. MPN-HBP1 nanoparticles entered tumor cells efficiently and innocuously, induced ferroptosis, and inhibited the malignant proliferation of tumors by regulating the HBP1-UHRF1-CDO1 axis. This study provides a new perspective for further research on the regulatory mechanism underlying ferroptosis and its potential role in tumor therapy.

## Introduction

Ferroptosis is an iron-dependent form of regulatory cell death that is caused by the loss of cellular redox homeostasis, thus to uncontrolled lipid peroxidation and eventually cell death [1]. Ferroptosis is associated with ischemia-induced pathological cell death and different types of cancer [2]. Many types of tumor cells are susceptible to ferroptosis after drug treatment, including cervical cancer, hepatocellular carcinoma, pancreatic cancer, and renal cell carcinoma [3]. Therefore, inducing ferroptosis in tumor cells has become an important aspect of tumor therapy. However, the regulatory mechanisms underlying ferroptosis have yet to be fully elucidated.

Human cysteine dioxygenase 1 (CDO1), an enzyme that adds molecular oxygen to the sulfur of cysteine and converts the thiol to a sulfinic acid known as cysteine sulfinic acid (3-sulfinoalanine). CDO1-induced cysteine deficiency has been shown to reduce glutathione (GSH)

S2C and S4C are deposited in the Zenodo database (https://zenodo.org/record/7990834).

**Funding:** This work was supported by grants of the National Natural Science Foundation of China (No. 82073068, 81874141, and 81672717) and Beijing Natural Science Foundation Grant (No. 7212056) to XWZ. The funders had no role in study design, data collection and analysis, decision to publish, or preparation of the manuscript.

**Competing interests:** The authors have declared that no competing interests exist.

**Abbreviations:** ACSL4, acyl-CoA synthetase long chain family member 4; BAP1, BRCA1 associated protein 1; BSO, Butylamine-Sulfoximine-L; CDO1, cysteine dioxygenase 1; ChIP, chromatin immunoprecipitation; CTRP, Cancer Therapeutics Response Portal; DFO, Deferoxamine; DMEM, Dulbecco's Modified Eagle Medium; EDTA, ethylene diamine tetraacetic acid; FDA, Food and Drug Administration; Fer-1, ferrostatin-1; GPX4, glutathione peroxidase 4; GSEA, Gene Set Enrichment Analysis; GSH, glutathione; HBP1, HMG box-containing protein 1; HCC, hepatocellular carcinoma; IHC, immunohistochemistry; MDA, malondialdehyde; MPN, metal polyphenol network; MPO, myeloperoxidase; NAC, N-acetyl-l-cysteine; NADPH, nicotinamide adenine dinucleotide phosphate; Ner-1, necrostatin-1; NSCLC, non-small cell lung cancer; PEI, polyethyleneimine; ROS, reactive oxygen species; shRNA, short hairpin RNA; TEM, transmission electron microscopy; TFRC, transferrin receptor; UHRF1, ubiquitin-like with PHD and RING finger domains 1; 4-HNE, 4-hydroxynonenal.

synthesis and weaken the antioxidant capacity of cells, eventually leading to an increase in the levels of reactive oxygen species (ROS) and the induction of ferroptosis [4]. Research has shown that the inhibition of CDO1 expression contributes to the elevated synthesis of GSH as well as a reduction in ROS levels, ultimately resulting in resistance to Erastin-induced ferroptosis [5]. Thus, high expression levels of CDO1 is an important factor in the development of ferroptosis.

HMG box-containing protein 1 (HBP1), as a dual transcription factor, inhibits its target genes by directly binding to specific affinity elements, such as *N-MYC*, *C-MYC*, *p47phox*, *DNMT1*, *EZH2*, and *AFP* [6–11], all of which are oncogenes or genes that promote tumor development. HBP1 also transcriptionally activates several downstream genes, including *p16*, *p21*, *myeloperoxidase (MPO)*, and *histone H1* [12–15], which are all tumor suppressor genes. HBP1 regulates the cell cycle and inhibits cell proliferation by regulating the expression of its downstream cell cycle regulatory factors, thus inhibiting the occurrence and development of tumors. However, it remains unknown as to whether HBP1, as an important regulator of the cell cycle and metabolism, is involved in ferroptosis and whether it plays a tumor suppressor role by regulating ferroptosis. Ubiquitin-like with PHD and RING finger domains 1(UHRF1) protein is an epigenetic modification factor that plays a significant role in DNA methylation and histone methylation. UHRF1 is expressed at high levels in various malignant tumors, including breast, bladder, and prostate cancer, and is involved in the occurrence and progression of tumors [16–18]. In addition, UHRF1 can inhibit cell apoptosis via ROS-related signaling pathways in gastric cancer [19] and enhance the invasiveness of tumor cells via the Keap1-Nrf2 pathway in pancreatic cancer [20]. Therefore, UHRF1 may represent a crucial regulatory factor in tumorigenesis and development; however, the regulatory mechanisms upstream of UHRF1, especially in terms of transcriptional regulation, remain unclear.

Over recent years, nanomaterials have been widely used in laboratory research and clinical practice [21,22]. Some researchers have used nanoparticles synthesized by a metal polyphenol network (MPN) approach as carriers for inducers of ferroptosis such as Erastin or iron-based nanomaterials such as $Fe_3O_4$ nanoparticles to accurately induce ferroptosis in tumor tissues, so as to achieve therapeutic action [23–25]. The combination of induced ferroptosis and nanotechnology enhances the stability, biosafety, and targeting of drugs in vivo.

In this study, we determined that HBP1 up-regulates the expression of CDO1 at the epigenetic level by inhibiting the expression of the *UHRF1* gene at the transcription level, thereby promoting CDO1-mediated ferroptosis in tumor cells. Based on our findings, we combined tannic acid, a food additive extracted from green tea and approved by the Food and Drug Administration (FDA), with $Fe^{3+}$ to form MPN on the surface of a polyethylenimine-HBP1 plasmid complex (PEI-HBP1). We found that MPN-HBP1 nanoparticles enhanced the induction of ferroptosis. MPN-HBP1 was internalized by tumor cells and introduced a significant amount of $Fe^{3+}$ into the cells to induce the Fenton reaction to produce ROS, thus resulting in serious lipid peroxidation in the biomembrane. The exogenous expression of HBP1 enhanced ferroptosis in tumor cells by regulating the UHRF1-CDO1 signaling pathway. Herein, we evaluated the efficacy of MPN-HBP1 as a novel nanodrug with the capacity to induce ferroptosis in cancer therapy. This strategy may provide an option for improving the outcome of traditional cancer therapy.

## Results

### HBP1 regulates UHRF1 expression

Previously, we reported that HBP1-mediated transcriptional regulation of the methyltransferase DNMT1 induces global DNA hypomethylation during cell senescence [8]. We also

identified HBP1 as an intriguing and important factor in regulating the methylation state of DNA. UHRF1 regulates DNA methylation at the epigenetic level by recruiting DNMT1 and links DNA methylation and methylation maintenance following cell division [26–28]. Therefore, in the present study, we investigated whether UHRF1 is another target of HBP1.

First, we evaluated the expression levels of HBP1 and UHRF1 in pathological sections taken from various cancer patients (hepatocellular carcinoma, lung adenocarcinoma, and colon adenocarcinoma) by immunohistochemistry (IHC) staining. We observed a statistically significant inverse correlation between the protein levels of HBP1 and UHRF1 in the 3 types of tumors with high expression levels of UHRF1 (Fig 1A and 1B). Accordingly, the expression levels of HBP1 mRNA were negatively correlated with that of UHRF1 in TCGA public databases for liver, breast, and lung cancers (S1A Fig).

To determine whether HBP1 is a repressor of the *UHRF1* gene, we overexpressed HBP1 in HeLa, HepG2, and Huh7 cells (Fig 1C). The levels of UHRF1 protein (left panel) and mRNA

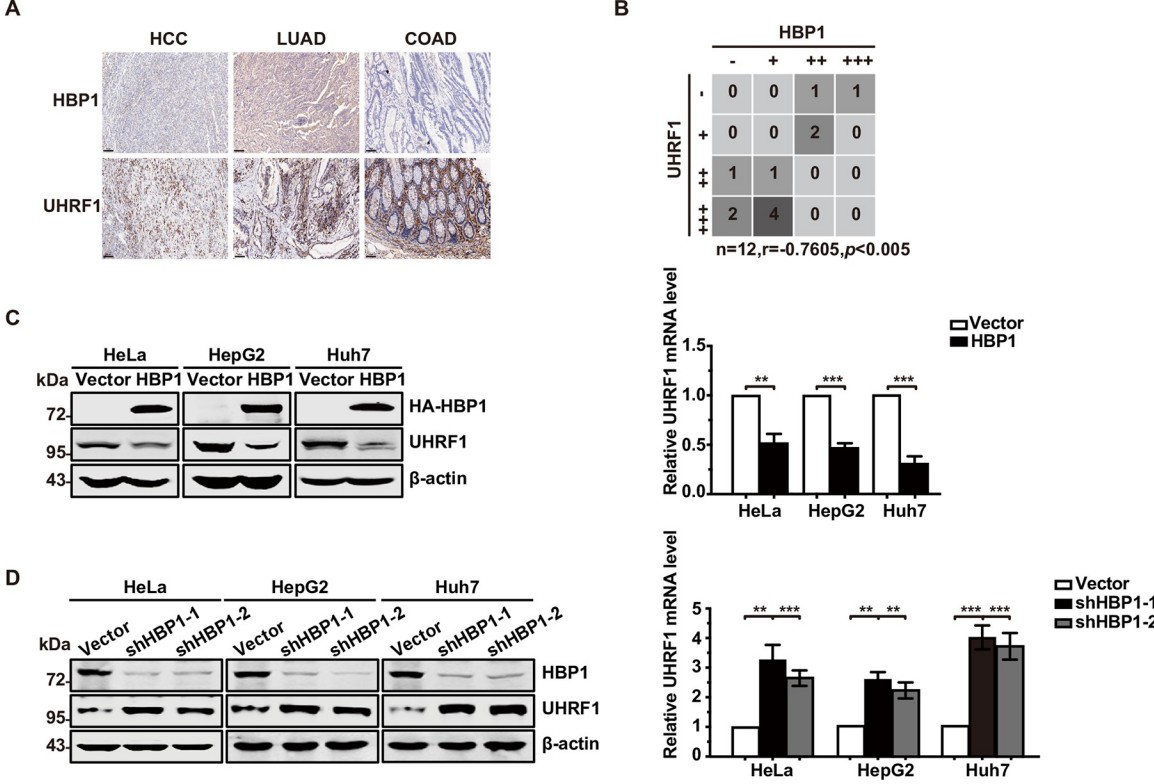

**Fig 1. HBP1 regulates UHRF1 expression.** (A) IHC staining of HBP1 and UHRF1 in pathological sections of cancer patients (*n* = 12/group). Scale bar = 100 μm. (B) Two-tailed Pearson correlation analysis method was used to calculate the correlation between the expression of HBP1 and UHRF1. (C) HBP1 overexpression decreases UHRF1 protein and mRNA expression. The protein levels of HBP1 and UHRF1 in cell lysate was measured by western blotting in HeLa, HepG2, and Huh7 cells transfected with pCDNA3-HBP1 or pCDNA3 (as a control). β-actin was used as a control, respectively (left panel). The mRNA level of UHRF1 was measured by real-time PCR in HeLa, HepG2, and Huh7 cells transfected with pCDNA3-HBP1 or pCDNA3 (right panel). (D) HBP1 knockdown by shRNA increases UHRF1 protein and mRNA expression. The protein levels of HBP1 and UHRF1 in cell lysate was measured by western blotting in HeLa, HepG2, and Huh7 cells stably transfected with pLL3.7-shHBP1-1, pLL3.7-shHBP1-2, or pLL3.7 (as a control) through lentiviral infection. β-actin was used as a control, respectively (left panel). The mRNA level of UHRF1 was measured by real-time PCR in HeLa, HepG2, and Huh7 cells stably transfected with pLL3.7-shHBP1-1, pLL3.7-shHBP1-2, or pLL3.7 (right panel) through lentiviral infection. The underlying data for Fig 1B–1D can be found in S1 Data. Differences between 2 groups were calculated using a two-tailed Student *t* test. Error bars represent SD., *p* < 0.05, **, *p* < 0.01, ***, *p* < 0.001. COAD, colon adenocarcinoma; HBP1, HMG box-containing protein 1; HCC, hepatocellular carcinoma; IHC, immunohistochemical; LUAD, lung adenocarcinoma; shRNA, short hairpin RNA; UHRF1, ubiquitin-like with PHD and RING finger domains 1.

(right panel) were reduced by HBP1 overexpression in HeLa, HepG2, and Huh7 cells. To confirm the endogenous regulatory activity of HBP1 on UHRF1 expression, we knocked down the expression of HBP1 using short hairpin RNAs (shRNAs). As shown in Fig 1D, the knockdown of HBP1 resulted in increased levels of UHRF1 protein (left panel) and mRNA (right panel) in the 3 cell lines. These results suggest that HBP1 inhibits UHRF1 expression at the transcriptional level.

## HBP1 represses the *UHRF1* gene by binding to an affinity site in the *UHRF1* promoter

Next, we investigated whether HBP1 inhibits the transcriptional activity of the *UHRF1* promoter through sequence-specific DNA binding. We cotransfected HEK293T cells with distinct fragments of the *UHRF1* promoter (−1,783 to +74, −1,536 to +74, −1,285 to +74, −1,121 to +74 from the transcriptional start site) and HBP1. As shown in Fig 2A, HBP1 inhibited some *UHRF1* promoter fragments (−1,783 to +74, −1,536 to +74, and −1,285 to +74) but had no effect on one *UHRF1* fragment (−1,121 to +74), thus indicating that site of affinity for HBP1 was between −1,285 and −1,121 bp in the *UHRF1* promoter. To further verify the DNA binding requirement for the functional activity of HBP1, we constructed a deletion reporter for the *UHRF1* promoter, which abolished the HBP1 affinity site between −1,173 and −1,155 bp. HBP1 inhibited the activity of the wild-type *UHRF1* promoter but had no effect on the mutant promoter (Fig 2B). In order to investigate whether the transcriptional repression of HBP1 depends on DNA binding, we used 2 mutants of HBP1: pmHMG (which had 3 amino acid mutations in the HMG domain and lacked DNA binding ability) and DelEx7 (which was isolated from breast cancer tissue and lacked the DNA binding domain and part of the repression domain) [8,12]. As shown in Fig 2C and 2D, wild-type HBP1 reduced the activity of the *UHRF1* promoter and the level of protein while the overexpression of pmHMG and DelEx7 had no effect, indicating that HBP1 transcriptional inhibition of *UHRF1* gene expression depends on DNA binding. Since HBP1 inhibited the activity of the *UHRF1* promoter, we tested whether HBP1 binds directly to the *UHRF1* promoter. Chromatin immunoprecipitation (ChIP) assay (Fig 2E) demonstrated that HBP1 bound directly to the specific affinity site in the *UHRF1* promoter in vivo. In contrast, pmHMG and DelEx7 did not bind to the *UHRF1* promoter. Therefore, we concluded that HBP1 inhibits the *UHRF1* gene by binding to a site of affinity within the *UHRF1* promoter.

## HBP1 reduces cellular antioxidant capacity and therefore sensitizes tumor cells to ferroptosis

To investigate potential physiological or pathological biological processes involving HBP1, we first downloaded and analyzed relevant mRNA sequence data from the LIHC data set (Cbioportal) of TCGA database. Gene Set Enrichment Analysis (GSEA) identified HBP1 as generally associating with oxidative stress, cell death in response to oxidative stress, oxidoreductase activity, and lipid oxidation (S1B Fig). We specifically focused on oxidation stress process and then asked if HBP1 expression could be linked to the regulation of redox homoeostasis in tumor cells. The effect of HBP1 levels on redox balance in HepG2 cells was then explored using a HBP1 overexpression system and lentivirus-mediated knockdown. We first examined the subsequent effect on the nicotinamide adenine dinucleotide phosphate (NADPH) and GSH systems that represent important guardians for maintaining cell redox homoeostasis. HBP1 overexpression in HepG2 cells resulted in significantly decreased ratios of NADPH/NADP$^+$ and GSH/GSSG (S2A and S2B Fig, left panels). In contrast, HBP1 knockdown in HepG2 cells led to an increase in the ratios (S2A and S2B Fig, right panels), suggesting that

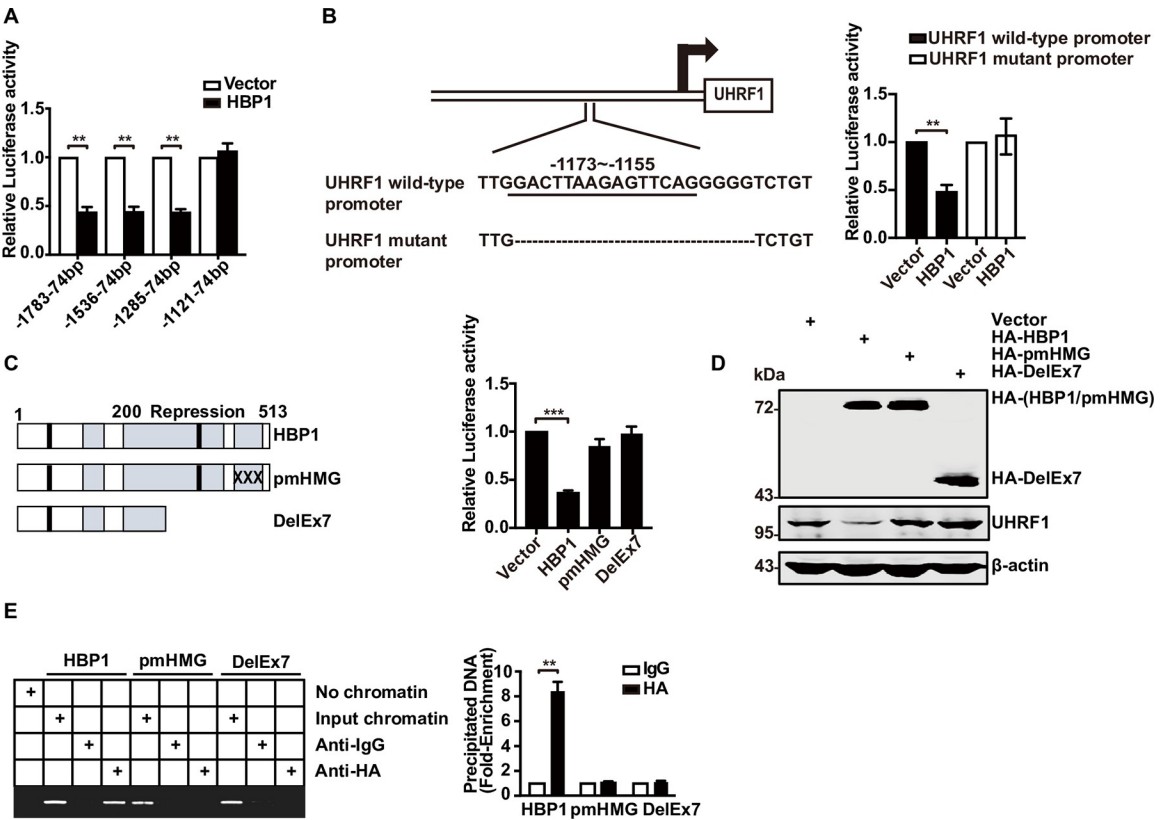

**Fig 2. HBP1 represses *UHRF1* gene by binding an affinity site in the *UHRF1* promoter.** (A) Relative activity of HBP1 on the *UHRF1* promoters with various lengths. (B) The integrity of affinity site is indispensable for HBP1 suppressing *UHRF1* promoter in vivo. Shown is schematic diagram of the wild-type *UHRF1* promoter and its mutant promoter (left panel) and the relative activities of HBP1 on the wild-type *UHRF1* promoter and mutant *UHRF1* promoters (right panel). (C) Relative activities of HBP1 and associated mutants on the *UHRF1* promoter in cotransfected HEK293T cells. Shown is schematic diagram of wild-type HBP1 and associated mutants (left panel). Luciferase activity was determined after transfection (right panel). (D) Expression of exogenous HBP1 decreases UHRF1 protein level. HEK293T cells were transfected with HBP1 and associated mutants. The protein level was measured by western blotting. (E) HBP1 binding to the endogenous *UHRF1* promoter requires the HMG domain. ChIP assays were used to test the binding of exogenous HBP1 to endogenous *UHRF1* gene. HEK293T cells were transfected with HA-HBP1, HA-pmHMG, or HA-DelEx7. The region from position −1,289 to position −1,067 contains the HBP1 affinity site and was analyzed by specific PCR. Anti-HA antibody was used in the indicated lanes. The underlying data for Fig 2A, 2B, 2C and 2E can be found in S1 Data. Differences between 2 groups were calculated using a two-tailed Student *t* test. One-way ANOVA was performed to assess differences among multiple groups. Error bars represent S.D. *, $p < 0.05$, **, $p < 0.01$, ***, $p < 0.001$. ChIP, chromatin immunoprecipitation; HBP1, HMG box-containing protein 1; UHRF1, ubiquitin-like with PHD and RING finger domains 1.

HBP1 can alter the redox balance in HepG2 cells by transitioning to oxidation. To determine whether redox changes caused by HBP1 is accompanied by ROS accumulation, we tested intracellular ROS levels in the HepG2 cells with HBP1 overexpression or knockdown. Since endogenic ROS are mainly produced in mitochondria, we also tested mitochondrial ROS levels in these cells. We found that both intracellular ROS levels and mitochondrial ROS levels in HBP1 overexpressed cells were significantly elevated compared to the vector, and this pheno-type could be reversed by adding of the antioxidant N-acetyl-l-cysteine (NAC). In contrast, the intracellular total and mitochondrial ROS levels in HBP1 knockdown cells were significantly reduced compared with the vector (S2C and S2D Fig). We further examined potential changes in mitochondrial membrane potential (ΔΨm) in these cells to determine if mitochondrial function was impaired by the elevated ROS. Compared with the vector, HBP1 overexpressed cells had a higher proportion of depolarized mitochondria, showing a decrease in ΔΨm, while

HBP1 knockdown cells had an increase in $\Delta\Psi$m (S2E Fig). NAC treatment restored the decreased $\Delta\Psi$m associated with HBP1 overexpression in HepG2 cells. These results suggest that HBP1 raises ROS levels and thus damages $\Delta\Psi$m in HepG2 cells. We further hypothesized that HBP1 might restrict the antioxidant capacity of HepG2 cells, thereby sensitizing these cells to ROS damage. In support of this hypothesis, HBP1 overexpressed cells showed impaired cellular antioxidant capacity and HBP1 knockdown cells showed enhanced antioxidant capacity compared with the vectors (S2F Fig). We then treated HepG2 cells with hydrogen dioxide ($H_2O_2$) and found that HepG2 cells with HBP1 overexpression were more sensitive to $H_2O_2$ treatment than the vector, while HepG2 cells with HBP1 knockdown were more resistant to $H_2O_2$ treatment (S2G Fig). Taken together, these results strongly suggest that HBP1 overexpression impairs antioxidant capacity of HepG2 cells, disrupts redox balance, and sensitizes HepG2 cells to oxidative stress-induced damage. Then, we used the Cancer Therapeutics Response Portal (CTRP), which allowed us to analyze correlations between gene expression and the response to 481 compounds across different cancer cell lines. Liver cancer cell line data from the CTRP revealed a significant correlation between HBP1 expression and sensitivity to RSL-3 (S2H Fig), which targets glutathione peroxidase 4 (GPX4) to induce ferroptosis [29]. Our previous work has shown that HBP1 is involved in the biological process of $H_2O_2$-induced apoptosis [30]. Therefore, we planned to investigate whether HBP1 participates in ferroptosis, another form of cell death, by regulating UHRF1.

Next, we investigated the induction of ferroptosis by RSL-3 or Erastin, both of which are well-known inducers of ferroptosis. As shown in S2I Fig, HeLa and HepG2 cells were treated with Erastin or RSL-3. Cell viability decreased with the increase of Erastin or RSL-3 concentrations. Ferrostatin-1 (Fer-1), an inhibitor of ferroptosis, was able to rescue RSL-3- or Erastin-induced cell death (S2J Fig), thus confirming the induction of ferroptosis by RSL-3 or Erastin. To determine whether the HBP1-UHRF1 axis plays a pivotal role in regulating ferroptosis, we analyzed the protein levels of HBP1 and UHRF1 during ferroptosis in HeLa, HepG2, and Huh7 cells. Increasing concentrations of Erastin or RSL-3 led to an increase in HBP1 protein level and a reduction in UHRF1 protein level (Fig 3A and 3B, left panels), whereas Fer-1 rescued the Erastin/RSL-3-induced changes in protein levels (Fig 3C). We also introduced an iron scavenger, Deferoxamine (DFO), another ferroptosis inhibitor, which could rescue the Erastin-induced changes in HBP1 and UHRF1 protein levels as well (S2K Fig). However, the mRNA expression of HBP1 remained unchanged following Erastin or RSL-3 treatment (Fig 3A and 3B, right panels). These results suggest that the HBP1-UHRF1 axis may contribute to ferroptosis and that the stability of HBP1 protein may change during ferroptosis.

Subsequently, we used a selective 26S proteasomal inhibitor (MG-132) to address the effect of ubiquitination-mediated proteasomal degradation on HBP1 protein during ferroptosis. The levels of HBP1 protein were increased by treatment with MG-132. Erastin treatment did not induce a further increase in HBP1 (Fig 3D). As shown in Figs 3E and S3, Erastin clearly increased the half-life of HBP1 protein from 20.7 min to 53.5 min, suggesting that Erastin-induced increase in HBP1 protein level is mediated in a proteasome-dependent manner. To determine whether Erastin inhibits the ubiquitination and degradation of HBP1, we transfected HEK293T cells with HA-HBP1 with or without Erastin treatment and then exposed the cells to MG132 for 6 h. Subsequently, the cells were subjected to anti-HA immunoprecipitation followed by western blotting with an anti-ubiquitin antibody. As shown in Fig 3F, Erastin significantly inhibited HBP1 ubiquitination. Thus, we concluded that Erastin increases the expression of HBP1 protein by inhibiting HBP1 ubiquitination-mediated proteasomal degradation.

Analyses showed that HBP1 inhibited the protein level, mRNA level, and promoter activity of UHRF1 in the presence of Erastin in a more significant manner than with HBP1 alone

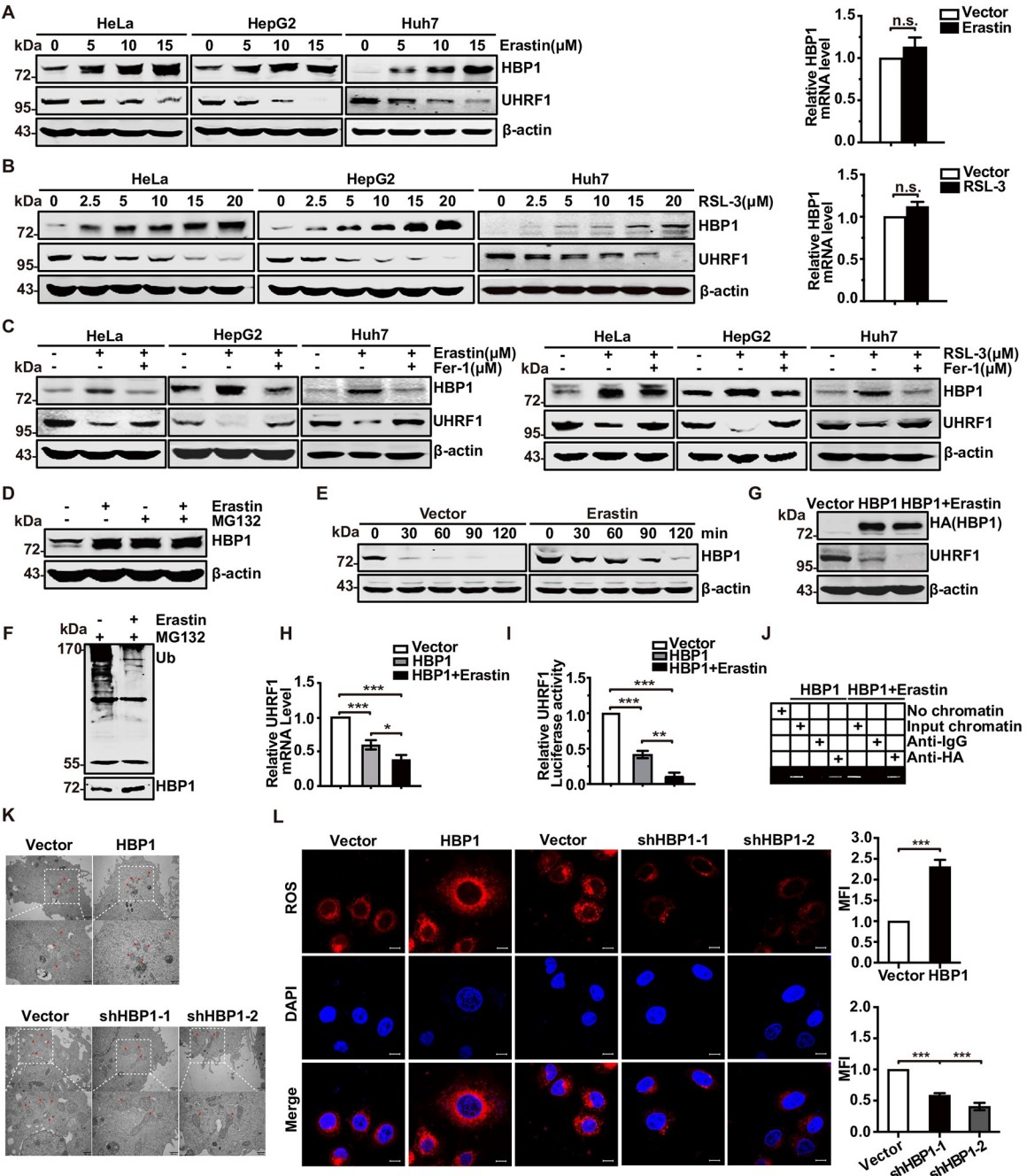

**Fig 3. HBP1 reduces cellular antioxidant capacity and therefore sensitizes tumor cells to ferroptosis.** (A) HBP1 protein increased and UHRF1 protein decreased during ferroptosis. After HeLa, HepG2, and Huh7 cells were treated with different concentrations of Erastin, the protein expression of HBP1 and UHRF1 was detected by western blotting (left panel) and the mRNA expression level of HBP1 was detected by qPCR after HeLa cells were treated with 10 μM Erastin (right panel). (B) After HeLa, HepG2, and Huh7 cells were treated with different concentrations of RSL-3, the protein expression of HBP1 and UHRF1 was detected by western (left panel) and the mRNA expression level of HBP1 was detected by qPCR after HeLa cells were treated with 5 μM RSL-3 (right panel). (C) Protein levels of HBP1 and UHRF1 in HeLa, HepG2, and Huh7 cells were treated with Erastin (10 μM, left panel)/RSL-3 (5 μM, right panel) alone or in combination with Fer-1 (10 μM). (D) Erastin inhibits HBP1 ubiquitination-mediated proteasome degradation. HeLa cells were treated with 10 μM Erastin with or without MG132; the protein level of HBP1 was measured by western blotting. (E) Erastin extends the half-life of HBP1 protein. HeLa cells were treated with Erastin for 24 h, and cells were incubated with the protein synthesis inhibitor CHX for 0, 30, 60, 90, or 120 min before collect. HBP1 protein levels were detected by western blotting. (F) HEK293T cells were transfected HA-HBP1 with or without 10 μM Erastin treatment for 24 h and then exposed to MG132 for another 6 h prior to lysis. HBP1 protein was then isolated by immunoprecipitation and analyzed by anti-Ub antibody. (G) Erastin promotes the suppression of HBP1 on UHRF1

protein. HeLa cells were transfected HBP1 with or without 10 μM Erastin treatment. The protein levels of HBP1 and UHRF1 were measured by western blotting. (H) Erastin promotes the suppression of HBP1 on UHRF1 mRNA. The mRNA level of UHRF1 was measured by qPCR in HeLa cells transfected HBP1 with or without 10 μM Erastin treatment. (I) Erastin promotes the suppression of HBP1 on *UHRF1* promoter. HEK293T cells were cotransfected with *UHRF1* promoter and HBP1 with or without 10 μM Erastin treatment. Luciferase activity was determined after transfection. (J) Erastin promotes the interaction between HBP1 and *UHRF1* promoter. HEK293T cells were transfected HA-HBP1 with or without 10 μM Erastin treatment. The region from position −1,289 to position −1,067 contains the HBP1 affinity site and was analyzed by specific PCR. (K) Representative TEM images of the mitochondrial morphology in HBP1-OE and HBP1-KD HeLa cells treated with 10 μM Erastin for 24 h. Red arrows indicate mitochondria. Scale bar = 1 μm (top)/500 nm (bottom). (L) Representative images (left) and quantification (right) of ROS level in HBP1-OE and HBP1-KD HeLa cells treated with 10 μM Erastin for 24 h. Scale bar = 10 μm. The underlying data for Fig 3A, 3B, 3H, 3I and 3L can be found in S1 Data. Differences between 2 groups were calculated using a two-tailed Student *t* test. One-way ANOVA was performed to assess differences among multiple groups. Error bars represent S.D. *, $p < 0.05$, **, $p < 0.01$, ***, $p < 0.001$. CHX, cycloheximide; Fer-1, ferrostatin-1; HBP1, HMG box-containing protein 1; KD, knockout; OE, overexpression; ROS, reactive oxygen species; TEM, transmission electron microscopy; UHRF1, ubiquitin-like with PHD and RING finger domains 1.

(Fig 3G–3I). ChIP assays further showed that Erastin enhanced the binding of HBP1 to the *UHRF1* promoter (Fig 3J). These data indicate that the up-regulation of HBP1 protein by Erastin further enhances the transcriptional inhibition of HBP1 on UHRF1 by enhancing the binding of HBP1 to the *UHRF1* promoter.

To further determine the role of HBP1 in ferroptosis, we used transmission electron microscopy (TEM) to detect morphological changes in HeLa cells with HBP1 overexpression or HBP1 knockdown treated with Erastin. The cells overexpressing HBP1 and treated with Erastin possessed smaller mitochondria, diminished or vanished mitochondria cristae, and condensed mitochondrial membrane densities; these were typical mitochondrial phenotypes of ferroptosis when compared to vector cells. In contrast, HBP1 knockdown alleviated the abnormalities of mitochondrial morphology and cell death induced by Erastin (Fig 3K). We also measured ROS levels after interference with HBP1 expression; ROS levels are known to be the primary cause of ferroptosis. Results showed that the overexpression of HBP1 led to a significant increase in ROS levels in Erastin-induced HeLa cells, whereas HBP1 knockdown reduced the levels of ROS (Fig 3L). In conclusion, these results suggest that HBP1 promotes ferroptosis in tumor cells and sensitizes tumor cells to ferroptosis.

## HBP1 sensitizes tumor cells to ferroptosis by inhibiting UHRF1 expression in tumor cells

Since the HBP1-UHRF1 axis may contribute to ferroptosis in tumor cells, we speculate that HBP1 may increase the sensitivity of tumor cells to ferroptosis by inhibiting UHRF1. We next investigated the effect of HBP1 on UHRF1 expression in the context of Erastin treatment. HBP1 knockdown recovered the reduced UHRF1 levels induced by Erastin treatment in HeLa, HepG2, and Huh7 cells (Fig 4A), thus indicating that the inhibitory effect of HBP1 on UHRF1 is involved in the ferroptosis process in tumor cells.

To determine the type of cell death associated with increased HBP1 expression and Erastin induction, we also evaluated the effect of ferroptosis inhibitor (Fer-1, DFO), necroptosis inhibitor Necrosulfonamide (Nec-1), apoptosis inhibitor Ac-DEVD-CHO (Apo), and autophagy inhibitor Chloroquine (Chq) on the viability of cells. As shown in Fig 4B, for all tested cells (HeLa/HBP1, HeLa/shHBP1, and their vector counterparts), only Fer-1 and DFO were able to protect cells from Erastin-induced cell death. Nec-1, Apo, and Chq had no significant effect on Erastin-induced cell death. The overexpression of HBP1 enhanced Erastin-induced cell death while HBP1 knockdown had the opposite effect. These data indicate that Erastin causes tumor cell death by inducing ferroptosis and that HBP1 may enhance the antitumor effect of Erastin by regulating ferroptosis.

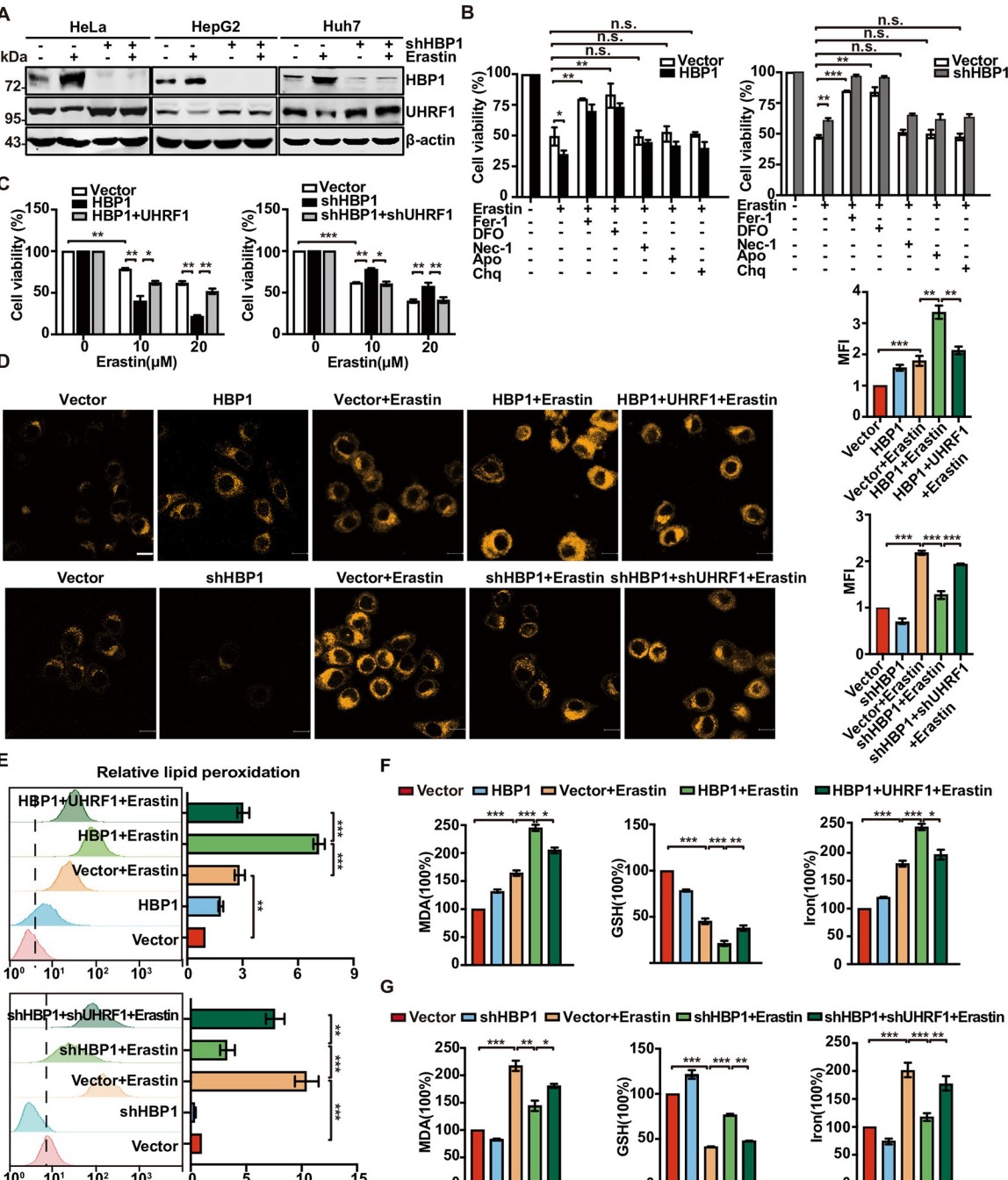

**Fig 4. HBP1 sensitizes tumor cells to ferroptosis by inhibiting UHRF1 expression in tumor cells.** (A) Protein levels of HBP1 and UHRF1 in HBP1 knockdown cells treated with 10 μM Erastin. (B) Indicated cells were treated with or without 10 μM Erastin for 24 h in the presence of different cell death inhibitors (Fer-1, 10 μM; DFO, 50 μM; Ac-DEVD-CHO, 59 μM; Necrosulfonamide, 20 μM; Chloroquine, 10 μM). Cell viability was measured using MTT. (C) Cell viability was conducted with HeLa cells stably transfected with vector, HBP1, HBP1+UHRF1 or vector, shHBP1, shHBP1+shUHRF1. (D-G) HBP1-UHRF1 axis sensitizes HeLa cells to ferroptosis. Indicated cells were treated with or without 10 μM Erastin for 24 h. (D) Cells were collected, and confocal was used to detect $Fe^{2+}$ levels. (E) Flow cytometry was used to detect lipid peroxides. (F, G) Indicated cells were lysed and MDA, GSH, and iron contents were measured. Scale bar = 10 μm. The underlying data for Fig 4B–4G can be found in S1 Data. Differences between 2 groups were calculated using a two-tailed Student *t* test. One-way ANOVA was performed to assess differences among multiple groups. Error bars represent SD. *, $p < 0.05$, **, $p < 0.01$, ***, $p < 0.001$. DFO, Deferoxamine; Fer-1, ferrostatin-1; GSH, glutathione; HBP1, HMG box-containing protein 1; MDA, malondialdehyde; UHRF1, ubiquitin-like with PHD and RING finger domains 1.

Reduced cell activity, increased intracellular $Fe^{2+}$, lipid peroxidation, and the depletion of GSH are recognized hallmarks of ferroptosis [31].To confirm whether HBP1 can promote Erastin-induced ferroptosis via UHRF1 down-regulation, we measured the levels of these markers in tumor cells transfected with HBP1, HBP1+UHRF1, HBP1shRNA, or HBP1shRNA+UHRF1shRNA following the administration of Erastin. We observed a reduction of cell viability following Erastin treatment in HeLa vector cells. The overexpression of HBP1, however, resulted in a further reduction in cell viability that could be restored by the overexpression of UHRF1. Conversely, the knockdown of HBP1 partially blocked the reduction of cell viability after Erastin treatment; this effect was reversed by knocking down UHRF1 (Fig 4C).

Labile iron ($Fe^{2+}$) is an important source of hydroxyl radical formation and can initiate lipid peroxidation via the Fenton reaction [32]. We found that both the overexpression and knockdown of HBP1 had a slight effect on the intracellular $Fe^{2+}$ content of HeLa cells (Fig 4D). However, the combination of HBP1 overexpression and Erastin treatment led to an obvious increase in $Fe^{2+}$ production than that seen with Erastin treatment alone in HeLa cells. This effect of HBP1 was restored by the overexpression of UHRF1. In contrast, the combination of HBP1 knockdown and Erastin treatment resulted in reduced $Fe^{2+}$ production when compared with Erastin treatment alone; this effect was reversed by the knockdown of UHRF1. These data suggest that HBP1 may account for $Fe^{2+}$ production or iron metabolism in ferroptosis by down-regulating UHRF1.

Furthermore, the overexpression of HBP1 led to a further increase in Erastin-induced lipid peroxidation, while HBP1 knockdown restrained the increase of lipid peroxidation induced by Erastin in HeLa cells. These actions of HBP1 overexpression and knockdown could be reversed by the overexpression and knockdown of UHRF1, respectively (Fig 4E). Malondialdehyde (MDA), a product of lipid peroxidation, another sensitive indicator of ferroptosis, is consistent with the level of lipid peroxidation (Fig 4F and 4G, left panels). In addition, GSH was significantly reduced after Erastin treatment. The overexpression of HBP1, however, resulted in a further reduction of GSH levels, and overexpression of UHRF1 restored this effect (Fig 4F, middle panel). HBP1 knockdown partially blocked the depletion of GSH arising from Erastin treatment. This effect could be reversed by the knockdown of UHRF1 (Fig 4G, middle panel). We also used a colorimetric method to detect the intracellular $Fe^{2+}$ content (Fig 4F and 4G, right panels), and the results were the same as those of fluorescent probe staining shown in Fig 4D. Collectively, these results suggest that HBP1 sensitizes tumor cells to ferroptosis by inhibiting UHRF1 expression.

Sorafenib is a first-line molecular-target drug for advanced hepatocellular carcinoma (HCC). The main function of sorafenib is to inhibit the proliferation and promote the apoptosis of HCC cells. However, the resistance of HCC cells to apoptosis makes hepatoma patients prone to drug resistance to sorafenib. Recently, sorafenib was identified as an inducer of ferroptosis [33]. The classic phenotype of ferroptosis, such as GSH depletion, elevated $Fe^{2+}$ and increased ROS production, is often observed in the cells treated with sorafenib. Therefore, promoting ferroptosis induced by sorafenib may be a new way to better treat HCC. According to the regulation of ferroptosis by HBP1-UHRF1 axis, we reasonably speculated that HBP1 might also increase the sensitivity of HCC cells to sorafenib by inhibiting the expression of UHRF1. To confirm this conjecture, we used Erastin or sorafenib to treat HepG2 cells transfected with HBP1, HBP1+UHRF1, shHBP1, shHBP1+shUHRF1. As shown in S4 Fig, similar to Erastin, solafenib treated HepG2 cells with HBP1 overexpression, resulting in a further decline in cell viability (S4A Fig), a more significant increase in intracellular $Fe^{2+}$, ROS, and MDA levels (S4B–S4D Fig), and a further decrease in GSH levels compared with solafenib treated alone (S4D Fig). Overexpression of UHRF1 reversed these effects of HBP1. However, solafenib treated HepG2 cells with HBP1 knockdown caused the opposite of these changes, and UHRF1

knockdown rescued these effects caused by HBP1 knockdown (S4A–S4C, S4E Fig). These data suggest that HBP1 also promotes the sensitivity of HCC cells to sorafenib-induced ferroptosis by inhibiting UHRF1 expression, thereby reducing the resistance of HCC cells to sorafenib.

## UHRF1 epigenetically regulates the methylation status of the *CDO1* promoter during ferroptosis

UHRF1 is the "core protein" of epigenetic regulation and has been shown to bind DNMT1 to hemi-methylated DNA to maintain the stability of methylated DNA. UHRF1 often represses the transcription of target genes by regulating DNA methylation of the promoter of target genes [34]. Therefore, we next investigated whether UHRF1 epigenetically regulates the expression of a specific marker gene for ferroptosis, thereby regulating the process of ferroptosis.

We overexpressed UHRF1 in HeLa cells to detect the mRNA expression of ferroptosis-related genes. As shown in Fig 5A, the mRNA expression of CDO1 decreased dramatically, while there were no significant changes in the expression of other mRNAs. In addition, the mRNA expression of CDO1 increased when UHRF1 was knocked down (Fig 5B), thus indicating that UHRF1 may epigenetically regulate the expression of the *CDO1* gene.

CDO1, a non–heme iron metalloenzyme, transforms cysteine to taurine by catalyzing the oxidation of cysteine to its sulfinic acid [35,36]. A deficiency of cellular cysteine reduces GSH synthesis and impairs cellular antioxidant capacity, ultimately resulting in enhanced ROS levels and the induction of ferroptosis. Therefore, studies have confirmed that the overexpression of CDO1 promotes the process of ferroptosis, thereby inhibiting tumorigenesis [5].

We confirmed that CDO1 protein levels were increased during Erastin-induced ferroptosis in HeLa, HepG2, and Huh7 cells (Fig 5C). Western blotting further showed that the overexpression of UHRF1 in HeLa, HepG2, and Huh7 cells led to significant reductions in CDO1 expression, as compared with vector cells (Fig 5D). The down-regulation of UHRF1 expression led to an evident increase in CDO1 expression (Fig 5E). Next, we investigated whether UHRF1 regulates the activity of the *CDO1* promoter in a manner that depended on DNA binding. Fig 5F showed that UHRF1 inhibited the activity of the *CDO1* promoter in a dose-dependent manner. The combination of UHRF1 overexpression and Erastin treatment partially reversed the inhibitory effect of UHRF1 alone on the *CDO1* promoter (Fig 5G). MSP assays were used to detect the effect of UHRF1 on the methylation of the *CDO1* promoter during ferroptosis. As shown in Fig 5H, UHRF1 overexpression increased the methylation of the *CDO1* promoter; the combination of UHRF1 overexpression and Erastin treatment reversed the increased methylation of the *CDO1* promoter induced by UHRF1. These data suggest that UHRF1 may inhibit ferroptosis by increasing the methylation level of the *CDO1* promoter region and by decreasing the expression level of CDO1.

Next, we investigated the role of UHRF1 hemi-mCpG and H3K9me2/3 binding activities in regulating methylation of the *CDO1* promoter. We constructed 2 mutants based on previous studies [37]. In one mutant, Tyr191 and Pro192 were changed to Ala (YP191/192AA); this mutant exhibited defective binding in H3K9me2/3. In the other mutant, Tyr478 and Thr479 were changed to Ala (YT478/479AA), thus abolishing hemi-mCpG-binding activity (Fig 5I). Analysis showed that the loss of H3K9me2/3 or hemi-mCpG binding activity in UHRF1 only partially affected the expression levels of mRNA and protein, or the activity of the *CDO1* promoter. However, the SRA/TDD double mutant of UHRF1 had no effect on *CDO1* mRNA and protein expression or promoter activity (Fig 5J–5L). These results suggest that UHRF1 methylation of the *CDO1* promoter depends on the simultaneous presence of its hemi-mCpG and H3K9me2/3 binding activities, thus indicating that UHRF1 plays a role in ferroptosis by regulating the methylation of the *CDO1* promoter.

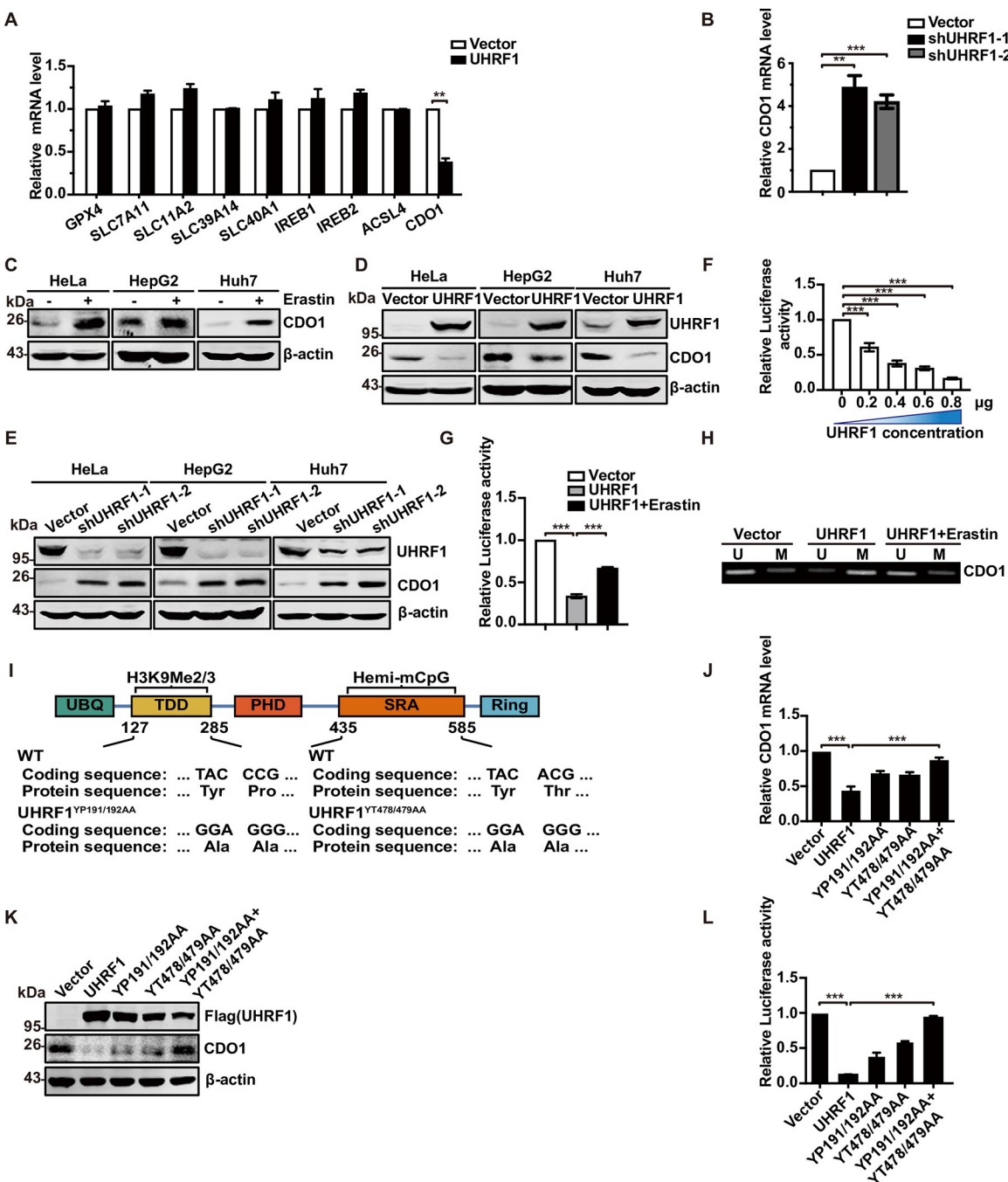

**Fig 5. UHRF1 epigenetically regulates methylation status of *CDO1* promoter during ferroptosis.** (A) Quantitative RT-PCR showing expression of ferroptosis-related genes in UHRF1 overexpressed cells. (B) The mRNA expression of CDO1 in UHRF1 knockdown cells. (C) The protein expression of CDO1 increased in Erastin-induced cells. Hela, HepG2, and Huh7 cells were treated with 10 μM Erastin for 24 h; cells were collected to detected using western blotting. (D) UHRF1 overexpression decreases CDO1 protein expression. The protein levels of UHRF1 and CDO1 in cell lysate were measured by western blotting in Hela, HepG2, and Huh7 cells transfected with pCDNA3.1UHRF1 or pCDNA3.1 (as a control). (E) UHRF1 knockdown by shRNA increases CDO1 protein expression. The protein levels of UHRF1 and CDO1 in cell lysate were measured by western blotting in Hela, HepG2, and Huh7 cells stably transfected with pLL3.7-shUHRF1-1, pLL3.7-shUHRF1-2, or pLL3.7 (as a control) through lentiviral infection. (F) UHRF1 suppresses *CDO1* promoter activity in a dose-dependent manner. Luciferase activity was detected in HEK293T cells cotransfected with the indicated reporter genes and UHRF1 plasmids. (G) Erastin promotes the suppression of UHRF1 on *CDO1* promoter. HEK293T cells were cotransfected with *CDO1* promoter and UHRF1 with or without 10 μM Erastin treatment. Luciferase activity was determined after transfection. (H) Erastin rescues UHRF1-induced specific *CDO1* promoter hypermethylation. Methylation levels of *CDO1* promoter were measured by MSP (right) in HeLa cells infected with pCDNA3.1 (as control) and pCDNA3.1-UHRF1. MSP using primer sets were listed in supplements.

(I) *CDO1*'s promoter methylation by UHRF1 depends on the simultaneous presence of its hemi-mCpG and H3K9me2/3 binding activities. Several mutants were constructed as shown. (J, K) The protein and mRNA levels of CDO1 were detected in HeLa cells transfected with vector, UHRF1, UHRF1$^{YP191/192AA}$, UHRF1$^{YT478/479AA}$, and UHRF1$^{YP191/192AA}$+UHRF1$^{YT478/479AA}$. (L) The occupation of UHRF1 on *CDO1* promoter in HEK293T cells was measured by luciferase assays. The underlying data for Fig 5A, 5B, 5F, 5G, 5J and 5L can be found in S1 Data. Differences between 2 groups were calculated using a two-tailed Student *t* test. One-way ANOVA was performed to assess differences among multiple groups. Error bars represent SD. *, $p < 0.05$, **, $p < 0.01$, ***, $p < 0.001$. CDO1, cysteine dioxygenase 1; MSP, methylation-specific PCR; RT-PCR, real-time PCR; shRNA, short hairpin RNA; UHRF1, ubiquitin-like with PHD and RING finger domains 1.

As a transcription factor, does HBP1 directly and specifically enhance *CDO1* gene transcription, thereby inducing ferroptosis? S5 Fig showed that although HBP1 increased CDO1 mRNA level and had no effect on the mRNA levels of other ferroptosis-related genes (S5A Fig), HBP1 did not directly bind to the *CDO1* promoter (S5B Fig), indicating that HBP1 transcriptionally enhance *CDO1* gene expression through UHRF1 in ferroptosis.

## The HBP1-UHRF1-CDO1 axis inhibits tumor cell proliferation and tumorigenesis

It was previously reported that CDO1 silencing promotes the proliferation of non-small cell lung cancer (NSCLC) by limiting the metabolism of cysteine to the wasteful and toxic by-products CSA and sulfite (SO$_3^{2-}$) and by depleting cellular NADPH [38]. To verify whether CDO1 inhibits the proliferation of HeLa, HepG2, and Huh7 cells, we constructed CDO1 knockout cells using the CRISPR/Cas9 system. We designed an sgRNA to target the third exon of the *CDO1* gene to prevent its transcription. *CDO1* knockout inhibited PTEN protein levels and increased the protein levels of p-AKT without affecting the total AKT protein level (S6A Fig). MTT and EdU assays showed that the cells in which CDO1 had been knocked out proliferated at a higher rate than vector cells (Fig 6A and 6B); the overexpression of CDO1 yielded opposite results (Figs 6C, 6D, and S6B). These results suggest that CDO1 inhibits the proliferation of tumor cells by regulating the PTEN-PI3K-AKT signaling pathway. Recently, some research showed that PI3K-AKT-mTOR signaling suppresses ferroptosis via SREBP-mediated lipogenesis [39]. Therefore, we speculate that CDO1 may promote ferroptosis by regulating PTEN-PI3K-AKT signaling pathway in addition to directly inhibiting glutathione synthesis. We added Erastin to induce ferroptosis in HeLa, HepG2, and Huh7 cells overexpressing CDO1 and also added PTEN inhibitor SF1670. The results showed that SF1670 rescued the decreased cell viability caused by overexpressing CDO1 (S6C Fig). We proved that CDO1 can partially promote ferroptosis through PTEN-AKT signal pathway.

Next, we investigated the effect of HBP1-UHRF1 axis on CDO1 growth inhibition. We detected expression of related proteins in PTEN/PI3K/AKT pathway in HeLa, HepG2, and Huh7. In addition, HBP1 overexpression increased the levels of CDO1 protein, thereby increasing the expression of PTEN protein and decreasing the levels of p-AKT level; in contrast, the coexpression of UHRF1 rescued the HBP1-induced changes in PTEN and p-AKT expression (Figs 6E and S6D). Furthermore, the knockdown of HBP1 reduced the expression of CDO1, thereby reducing PTEN levels and increasing the levels of p-AKT; however, there was no effect when UHRF1 was also knocked down (Figs 6F and S6E). Consistent with protein expression data, MTT and EdU assays showed that UHRF1 rescued the HBP1-induced reduction in cell proliferation. HBP1 knockdown cells grew at a faster rate, while UHRF1 knockdown rescued the high growth rate induced by HBP1 knockdown (Fig 6G and 6H). These data suggest that the HBP1-UHRF1-CDO1 axis inhibits tumor cell proliferation via the PTEN/PI3K/AKT signaling pathway.

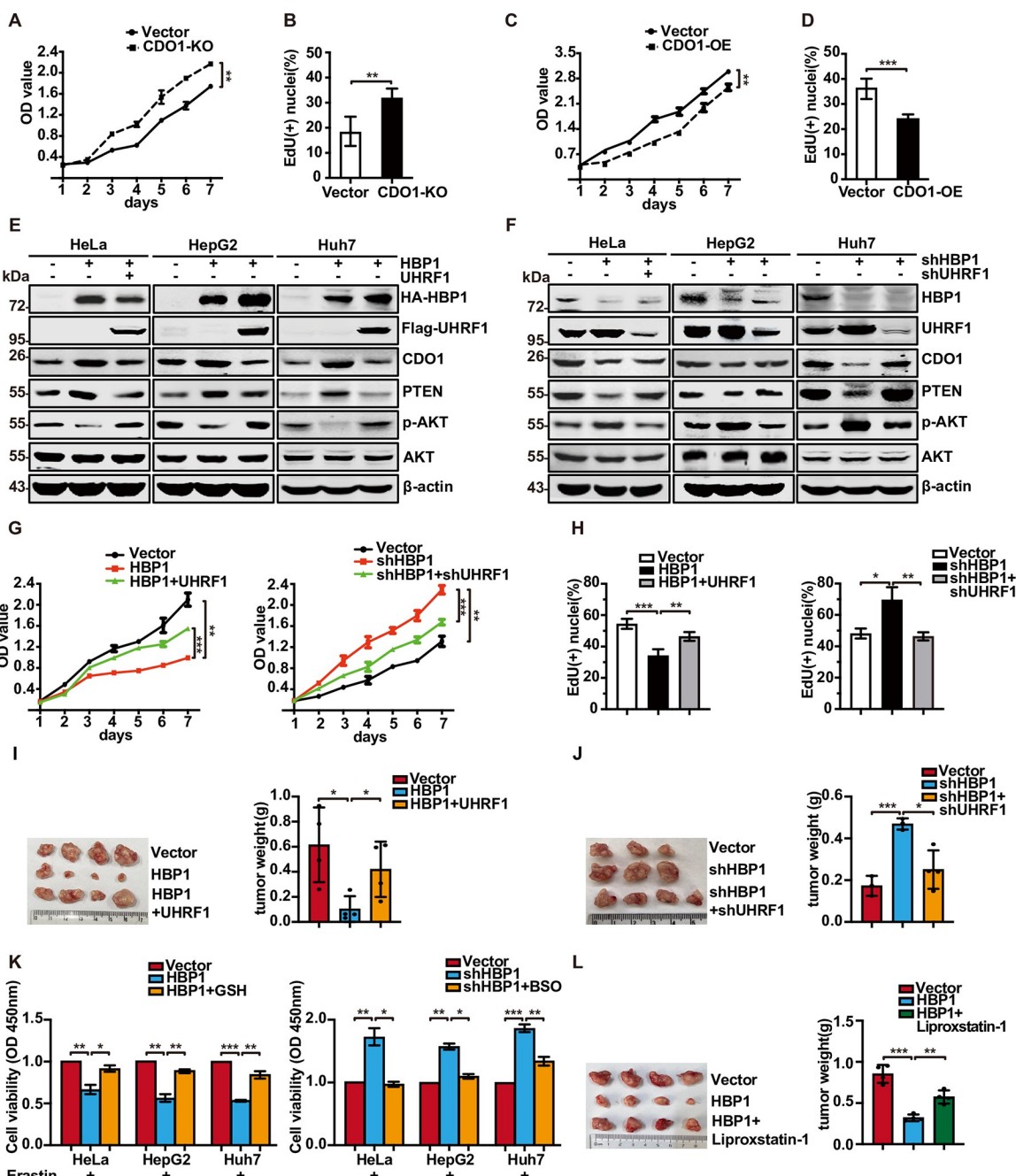

**Fig 6. The HBP1-UHRF1-CDO1 axis inhibits tumor cell proliferation and tumorigenesis.** (A, B) CDO1 knockout promotes cell proliferation. Cells with CDO1 knockout were analyzed by MTT assay (A) and EdU assay (B). (C, D) CDO1 overexpression inhibits cell proliferation. Cells with CDO1 overexpression were analyzed by MTT assay (C) and EdU assay (D). (E, F) Stable transfected HeLa, HepG2, and Huh7 cells were analyzed by western blotting. β-actin was detected as a control. (G) HeLa cells were stably transfected with vector, HBP1, HBP1+UHRF1 or vector, shHBP1, shHBP1+shUHRF1, and growth rates of cells were measured by MTT assay. (H) Cell proliferation was examined by EdU incorporation assay. (I, J) HepG2 cells stably transfected with vector, HBP1, HBP1+UHRF1 or vector, shHBP1, shHBP1+shUHRF1 were subcutaneously injected into nude mice. The tumors were weighed, 4 weeks after injection. (K) GSH synthesis is the key to mediate HBP1-regulated ferroptosis. HeLa, HepG2, and Huh7 cells with HBP1 overexpression treated with Erastin (10 μM) and GSH (10 μM) for 24 h. HeLa, HepG2, and Huh7 cells with HBP1 knockdown treated with Erastin (10 μM) and BSO (10 μM) for 24 h. Cell viability was measured by MTT. (L) HBP1 inhibits tumor proliferation by inducing ferroptosis. HepG2 cells stably transfected with vector and HBP1 were subcutaneously injected into nude mice. Mice were injected daily with Liproxstatin-1 (10 mg/kg, IP) during the course of the experiment. The tumors were weighed, 4 weeks after injection. The underlying data for Fig 6A–6D, 6G, and 6H–6L can be found in S1 Data. Data were analyzed using a two-tailed, unpaired Student $t$ test. One-way ANOVA was performed to assess differences among multiple groups. *, $p < 0.05$, **, $p < 0.01$, ***, $p < 0.001$. BSO, Butylamine-Sulfoximine-L;

CDO1, cysteine dioxygenase 1; GSH, glutathione; HBP1, HMG box-containing protein 1; IP, intraperitoneal; UHRF1, ubiquitin-like with PHD and RING finger domains 1.

To investigate the role of the HBP1-UHRF1-CDO1 axis during tumorigenesis in vivo, we subcutaneously injected 4-week-old BALB/c nude mice with HepG2 cells that stably expressed vector, HBP1 or HBP1+UHRF1, along with vector, shHBP1 or shHBP1+shUHRF1 individually. We found that UHRF1 reversed HBP1-induced tumor volume reduction (Fig 6I). HBP1 knockdown promoted the tumorigenic behavior of HepG2 cells; this was abolished by UHRF1 knockdown in HBP1 knockdown cells (Fig 6J).

Next, we explored whether HBP1-UHRF1-CDO1 axis can inhibit tumor proliferation by inducing ferroptosis. Because CDO1 mainly promotes ferroptosis by inhibiting the synthesis of glutathione, we added Erastin to HeLa, HepG2, and Huh7 cells overexpressing HBP1 and then added exogenous GSH. After 24 h, the cell viability was tested. The results showed that the exogenous GSH supplementation reversed the decline of cell viability caused by HBP1 (Fig 6K, left panel). Then, we added Erastin to HeLa, HepG2, and Huh7 cells with HBP1 knockdown and then added Butylamine-Sulfoximine-L (BSO) to inhibit glutathione synthesis. Finally, we detected the cell viability. The results showed that BSO reversed the increase in cell viability caused by HBP1 knockdown (Fig 6K, right panel). In addition, we constructed HepG2 cell lines that simultaneously knocks out CDO1 and overexpresses HBP1 to detect its cell viability. The results showed that HBP1 enhanced Erastin-induced ferroptosis, and the induced ferroptosis could be rescued by Fer-1. However, when CDO1 was knocked out, the cell viability was mostly restored (S6F Fig) but not completely recovered, indicating that HBP1 mainly induces ferroptosis through CDO1 but may also induce a small part of ferroptosis through other ways. Taken together, these results suggest that HBP1 inhibits cell viability primarily by inhibiting glutathione synthesis.

We further verified the above results in mice. We injected HepG2 cells stably overexpressing HBP1 subcutaneously in nude mice. After tumorigenesis, we injected ferroptosis inhibitor Liproxstatin-1 intraperitoneally into nude mice for treatment. The results showed that Liproxstatin-1 reversed the effect of HBP1 on tumor proliferation (Fig 6L). These results suggest that HBP1-UHRF1-CDO1 inhibits tumor growth by activating ferroptosis.

Collectively, our results suggest that the HBP1-UHRF1-CDO1 axis inhibits tumor cell proliferation and tumorigenesis.

## The nanoparticle MPN-HBP1 was designed to kill cancer cells via the ferroptosis pathway

Iron-based nanomaterials, such as superparamagnetic iron oxide nanoparticles, iron nanometal glasses, and iron-based metal organic frameworks, have been widely used as anticancer agents by virtue of their ability to induce ferroptosis. The ability of these materials to induce the generation of ROS is considered to play a key role in yielding satisfactory therapeutic effects [25,40]. However, the use of toxic materials and cumbersome synthetic routes significantly limit the use of these materials in biomedical applications. In recent years, MPNs have attracted much attention because of their easy "one-pot" synthesis, low cost, high biocompatibility, and wide range of potential applications [23,41].

Inspired from these previous findings, we used tannic acid, an FDA-approved food additive extracted from green tea, combined with ferric ions to form MPN on the surface of the polyethylenimine-HBP1 plasmid complex (PEI-HBP1), thus generating MPN-HBP1; the MPN coating was used to enhance the ferroptosis-inducing ability of HBP1.

MPN-HBP1 was synthesized for anticancer therapy according to the procedure shown in Fig 7A. First, the HBP1 plasmid was complexed with PEI via electrostatic interaction to form PEI-HBP1 complexes. The obtained spherical PEI-HBP1 complexes had a mean diameter of 40 nm, as determined by TEM. Then, ferric chloride and polyphenol were successively added into the PEI-HBP1 solution to prepare MPN-HBP1. Following assembly, the mean diameter of MPN-HBP1 increased to 65 nm (Fig 7B). Previous studies confirmed that the MPN membrane is a form of supramolecular network based on coordination between $Fe^{3+}$ and tannic acid [23]. To prove this, the driving force of MPN-HBP1 formation was further investigated by adding ethylene diamine tetraacetic acid (EDTA), NaCl, urea, and Tween 20 solutions, respectively, into MPN-HBP1. Immediately after the mixing of EDTA and MPN-HBP1, the color of the solution was bleached (S7A Fig). However, NaCl (the elimination of electrostatic forces), urea (the deformation of hydrogen bonds), and Tween 20 (the elimination of hydrophobic forces) were ineffective. These results indicated that coordination bonding is the dominant interaction during the formation of MPN coating.

After incubating MPN-HBP1 with HeLa cells for 24 h, there was a significant reduction in cell viability; this was consistent with Erastin treatment. Treatment with the MPN-vector or PEI-HBP1 also resulted in a slight reduction in cell viability (Fig 7C). These data suggest that MPN-HBP1 may induce the death of HeLa cells by activating ferroptosis, thus leading to a decline in cell viability. To confirm the type of cell death, different inhibitors of pathways associated with targeted cell death were applied to modulate the activity of cells that were treated with MPN-HBP1. As shown in Fig 7D, only Fer-1 significantly prevented MPN-HBP1-induced cell death; in contrast, Nec-1, Apo, and Chq were not able to rescue HeLa cells from MPN-HBP1-induced death. In addition, MPN-HBP1 significantly reduced the protein levels of UHRF1 and increased the protein levels of CDO1 (Fig 7E). Some protein changes have been proved to be biochemical markers of ferroptosis. The expression levels of GPX4, a scavenger of the peroxides and hydroxyl radicals that induce ferroptosis, is reported to be reduced during ferroptosis [42]. There are 2 other markers that promote ferroptosis, ACSl4 (acyl-CoA synthetase long chain family member 4) and TFRC (transferrin receptor). We found that MPN-HBP1 reduced the protein levels of GPX4 and increased the protein levels of TFRC and ACSl4 when compared with PEI-HBP1 (S7B Fig). We also determined the lipid oxidation efficacy in HeLa cells treated with PBS, PEI-HBP1, and MPN-HBP1 and found that the levels of the lipid peroxidation sensor DCFH-DA were highest in cells that were treated with MPN-HBP1 (Fig 7F). These results illustrated that MPN-HBP1 can inhibit cell viability by activating ferroptosis in a stronger manner than PEI-HBP1. MPN coating endowed the PEI-HBP1 complexes with the ability to induce ferroptosis. Our data support the fact that ferroptosis may play a dominant role in cell death regulated by MPN-HBP1.

Internalized nanoparticles are transported to lysosomes, and once intercepted, they are digested quickly. Therefore, the avoidance of lysosomes is a prerequisite for successful transfection. To explore whether MPN-HBP1 had a higher transfection efficiency than PEI-HBP1, we employed a lysosomal escape assay. As shown in Fig 7G, the cell nucleus, lysosome, and fluorescein-HBP1 of transfected HeLa cells were stained at different time point after transfection. At 1 h, negligible fluorescein-HBP1 entered cells in PEI-HBP1-treated cells and MPN-HBP1-treated cells. At 2 h, a small amount of fluorescein-HBP1 entered the MPN-HBP1-treated cells, while abundant fluorescein-HBP1 was entrapped in lysosomes of PEI-HBP1-treated cells. At 4 h, abundant fluorescein-HBP1 was still entrapped in lysosome of PEI-HBP1-treated cells, whereas more fluorescein-HBP1 had escaped from lysosomes and some even located in the nucleus in MPN-HBP1-treated cells. At 8 h, in PEI-HBP1-treated cells, HBP1 also escaped from lysosomes, but the fluorescence intensity was significantly reduced, indicating that a large number of HBP1 DNA was lysosomal degradation. Thus, we

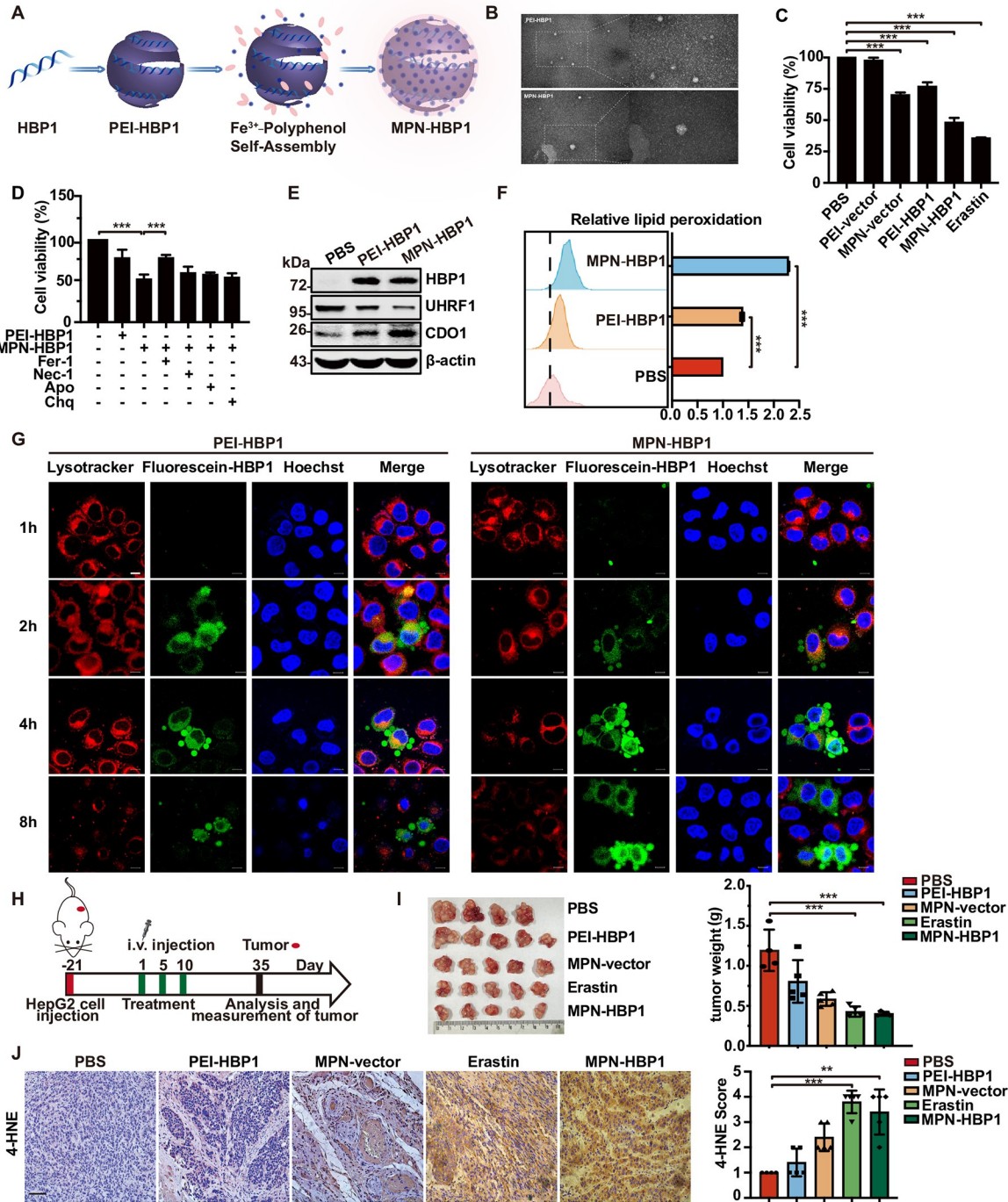

**Fig 7. The nanoparticles MPN-HBP1 was designed to kill cancer cells via ferroptosis pathway.** (A) Schematic illustration for the preparation of MPN-HBP1. (B) TEM images of PEI-HBP1 and MPN-HBP1. Scale bar = 100 nm (left)/50 nm (right). (C) Cell viability of HeLa cells after treated with PBS, PEI-vector, MPN-vector, PEI-HBP1, MPN-HBP1, and Erastin incubating for 24 h. (D) Cell viability of HeLa cells treated with PEI-HBP1 and MPN-HBP1 with or without Fer-1 (10 μM), Ner-1 (20 μM), Apo (59 μM), Chq (10 μM). (E) HBP1, UHRF1, and CDO1 protein levels of PBS-, PEI-HBP1-, and MPN-HBP1-treated HeLa cells. (F) Flow cytometry was used to detect lipid peroxides in HeLa cells after coincubation with PBS, PEI-HBP1, and MPN-HBP1 for 24 h. (G) Lysosome escape of HeLa cells treated with PEI-HBP1 and MPN-HBP1 at 1 h to 8 h by CLSM. Scale bar = 10 μm. (H) Schematic description of the experimental design used to establish the animal model. HepG2 cells were subcutaneously injected into nude mice. After the tumor volume had reached to 100 mm³, mice were randomized divided into different groups with 5 mice each. Then, mice were IV injected with 100 μL of PBS, PEI-HBP1, MPN-vector, Erastin, and MPN-HBP1 at first, fifth, and 10th days. Erastin was given at a dose of 5 mg/kg, and DNA in PEI-HBP1, MPN-vector, and MPN-HBP1 was given at a dose of 0.5 mg/kg ($n$ = 5 for all groups) in HepG2 tumor-bearing mice. (I) MPN-HBP1 inhibits tumorigenesis of HepG2 cells in nude mice. The tumors were weighed, 35 days after injection. (J) The expression of

4-HNE-positive cells ratio of subcutaneous tumor in indicated mice using IHC assay. Scale bar = 200 μm. The underlying data for Fig 7C, 7D, 7F, 7I and 7J can be found in S1 Data. Unpaired *t* test was used unless otherwise stated. One-way ANOVA was performed to assess differences among multiple groups. Error bars represent S.D. *, $p < 0.05$, **, $p < 0.01$, ***, $p < 0.001$. Apo, Ac-DEVD-CHO; CDO1, cysteine dioxygenase 1; Chq, Chloroquine; CLSM, confocal laser scanning microscope; Fer-1, ferrostatin-1; HBP1, HMG box-containing protein 1; IHC, immunohistochemistry; MPN, metal polyphenol network; Ner-2, Necrostatin-1; PEI, polyethylenimine; TEM, transmission electron microscopy; UHRF1, ubiquitin-like with PHD and RING finger domains 1; 4-HNE, 4-hydroxynonena.

concluded that MPN-HBP1 could avoid lysosomal degradation and function more effectively in cells.

Next, we investigated the efficiency of our nanoparticles to treat tumors; the treatment process is shown in Fig 7H. HepG2 cells were subcutaneously injected into BALB/c nude mice; subsequently, the volume of tumors increased to $100^3$ mm. PEI-HBP1, MPN-HBP1, MPN-vector, and Erastin were injected on days 1, 5, and 10. As depicted in Figs 7I and S7C, compared with the PEI-HBP1, MPN-vector, and PBS treatment group, treatment with MPN-HBP1 showed the most obvious inhibitory effect on tumor proliferation. Erastin, a ferroptosis-inducing drug, also exhibited satisfactory ability to inhibit tumors. However, this drug is associated with serious side effects, thus limiting its wider application. Body weight measurement (S7D Fig) and blood biochemistry analysis of liver and kidney functions (S7E Fig) showed that Erastin led to serious systemic toxicity. However, MPN-HBP1 treatment was associated with only negligible systemic toxicity. This may be attributed to the enhanced tumor targeting ability and reduced side effects of nanomedicine when compared with small molecule drugs. In addition, we investigated the expression of 4-hydroxynonenal (4-HNE), a lipid peroxidation product, in tumor tissues when treated with different particles (Fig 7J). The IHC staining score for 4-HNE in tumors treated with MPN-HBP1 and Erastin were significantly higher than that in other experimental groups, further indicating that MPN-HBP1 can inhibit tumor proliferation by inducing ferroptosis similar to Erastin. In conclusion, our MPN-HBP1 nanoparticles can effectively kill tumor cells by inducing ferroptosis, thus inhibiting the growth of tumors.

## Discussion

Ferroptosis plays an important regulatory role in the occurrence and development of tumors. Activating the ferroptosis pathway to alleviate tumor progression provides a promising strategy for tumor therapy. However, the functional changes and specific molecular mechanisms of ferroptosis still need to be fully elucidated. Over recent years, nanoparticle systems have been widely developed to overcome the obstacles associated with tumor therapy due to their high bioavailability and low toxicity. In this study, we identify HBP1 acting as a cellular pro-oxidant, specifically in the context of Erastin or sorafenib treatment of tumor cells. HBP1 disrupts redox homoeostasis and sensitizes tumor cells to ferroptosis by inhibiting activation of the *UHRF1* gene, an epigenetic factor, and down-regulates the level of UHRF1 protein. Reduced levels of UHRF1 are known to decrease the methylation level of the *CDO1* promoter, promoting the expression of CDO1 protein, and thus inhibiting the production of GSH, promoting the production of lipid peroxide, reducing the antioxidant capacity of tumor cells, and increasing the sensitivity of tumor cells to ferroptosis. We verified the regulatory role of the HBP1-UHRF1-CDO1 signaling pathway in ferroptosis. To enhance the effect of tumor therapy, we combined nanoparticles with HBP1 DNA to generate MPN-HBP1 nanodrugs. In addition to the high efficiency and low toxicity of nanodrugs, the tannic acid and ferric ions in the MPN coating enhanced the ability of HBP1 to induce ferroptosis in tumor cells. Our results are summarized in Fig 8. Collectively, our data revealed that MPN-HBP1 nanoparticles activate ferroptosis in tumor cells by regulating the HBP1-UHRF1-CDO1 signaling pathway, thus inhibiting the malignant proliferation of tumors.

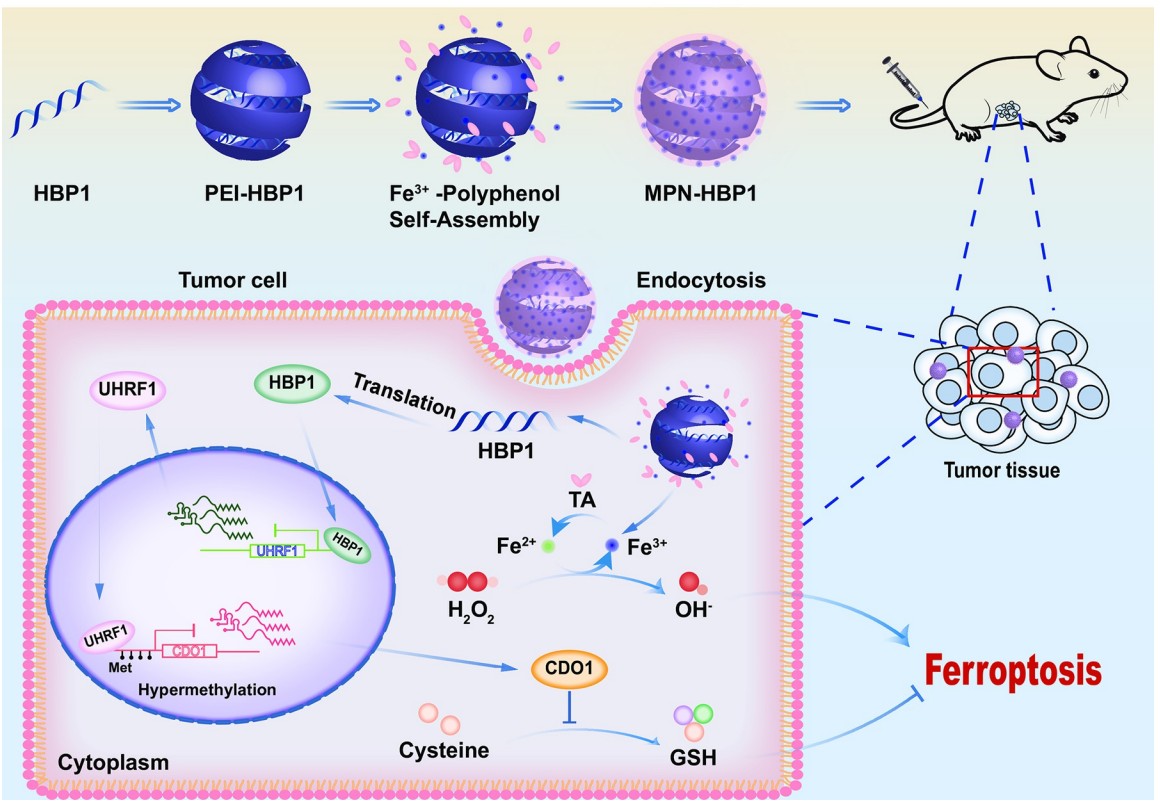

**Fig 8. Schematic illustration for MPN-HBP1-mediated anticancer therapy.** The PEI-HBP1 coordinates with tannic acid and ferric ions to form MPN-HBP1. After MPN-HBP1 nanoparticles reach the tumor site via passive targeting, the tannic acid and ferric ions in the MPN coating enhance the Fenton reaction to induce ferroptosis in tumor cells. Then, HBP1 transcriptionally inhibits the expression of the *UHRF1* gene and down-regulates the level of UHRF1 protein. Reduced levels of UHRF1 are known to decrease the methylation level of the *CDO1* promoter, promote the expression of CDO1 protein, and thus inhibit the production of GSH, promote the production of lipid peroxides, and induce ferroptosis. CDO1, cysteine dioxygenase 1; HBP1, HMG box-containing protein 1; MPN, metal polyphenol network; PEI, polyethyleneimine; UHRF1, ubiquitin-like with PHD and RING finger domains 1.

There is increasing evidence that transcription factors play an important role in regulating ferroptosis. These proteins can act as both promoters and blockers by regulating the expression of target genes involved in metabolic and antioxidant pathways. This complex transcriptional regulatory network exerts significant influence on the sensitivity to ferroptosis. The function of transcription factors in ferroptosis generally depends on the targeted genes. In our study, we found that the transcription factor HBP1 acts as an activator of ferroptosis and affects the expression of the ferroptosis-related gene *CDO1* via the transcriptional inhibition of UHRF1. Specifically, during ferroptosis, ubiquitination and the proteasomal degradation of HBP1 are both reduced, thus resulting in the accumulation of HBP1 protein. Increased levels of HBP1 enhance its accumulation on the *UHRF1* promoter and strengthens the transcriptional inhibition on UHRF1, thereby down-regulating the expression of UHRF1 and leading to an abnormal increase of CDO1. How does UHRF1 regulate the expression of CDO1? This involves the epigenetic regulation ability of UHRF1.

Epigenetic regulators, molecules that modify DNA without altering the genome, have emerged as a significant target and an effective option to modify the signaling pathways associated with ferroptosis, thus representing a novel and promising therapeutic potential target for ferroptosis. Over recent years, research has mainly focused on the epigenetic mechanisms of ferroptosis. Several epigenetic events have been identified in the regulation of ferroptosis. For

example, Zhang and colleagues found that tumor suppressor BRCA1 associated protein 1 (BAP1) reduced the occupancy of H2Aub on the *SLC7A11* promoter and inhibited the expression of SLC7A11 in a deubiquitination-dependent manner [43]. The antioxidant activity of SLC7A11 has been shown to inhibit ferroptosis. These authors further demonstrated that BAP1 inhibited cystine uptake by inhibiting the expression of SLC7A11, thus resulting in increased lipid peroxidation and ferroptosis. In addition, GPX4 has been shown to inhibit ferroptosis. Upstream of GPX4, there were few DNA methylation sites and the levels of H3K4me3 and H3K27ac were enhanced, thus suggesting that the high levels of GPX4 in cancer may arise from epigenetic regulation [44]. Therefore, it appears that epigenetic regulation plays a vital role in ferroptosis. UHRF1 has been widely investigated as a therapeutic target in cancers and is known to be a crucial factor linking histone H3 modification states, DNA methylation, and the regulation of tumorigenesis [45]. In the present study, we reveal a novel function of UHRF1 in ferroptosis and the determination of cell fate. Our data suggest that the levels of UHRF1 are significantly reduced in ferroptosis and that UHRF1 deficiency alters the DNA methylation status of the *CDO1* promoter, thus affecting the synthesis of GSH and finally promoting the progression of ferroptosis. Therefore, we believe that UHRF1 is a novel epigenetic regulator of ferroptosis.

In summary, we first found that HBP1 regulates the expression of UHRF1 at the transcriptional level and inhibits methylation of the *CDO1* promoter. CDO1 is involved in the biosynthesis of toxic cysteine into cysteine sulfate and mammalian taurine. Therefore, a deficiency of intracellular cysteine will reduce the synthesis of GSH, thus resulting in an increase in the levels of ROS and the induction of ferroptosis. Therefore, we confirm that the HBP1-UHRF1-CDO1 axis increases the sensitivity of tumor cells to ferroptosis and inhibits the proliferation of tumor cells and tumorigenesis. HBP1, as a tumor suppressor, is expressed at low levels in various cancers [8,11]. UHRF1 is a key epigenetic regulator that recruits DNMT1 to methylate DNA, thereby regulating the methylation of target genes and promoting the cell cycle and proliferation. UHRF1 is widely and highly expressed in a variety of cancers [17,18,20]. Previous reports have also described CDO1 as a tumor suppressor. CDO1 is usually silenced by promoter methylation during carcinogenesis. This is associated with distant metastasis and the poor prognosis of cancer [46–49]. On this basis, we hypothesize that increased levels of HBP1 expression activates ferroptosis and can contribute to tumor therapy via the HBP1-UHRF1-CDO1 axis. Therefore, we designed MPN-HBP1 nanoparticles to kill cancer cells via the ferroptosis pathway. Specifically, after cellular internalization, our nanoparticles induce cancer cells to release HBP1 and $Fe^{3+}$ through cascade reactions, thus creating therapeutic effect. Detailed mechanistic studies further showed that the HBP1-UHRF1-CDO1 axis significantly inhibits the glutathione synthesis pathway and produces an abundance of lipid peroxides. A significant influx of $Fe^{3+}$ into cells can induce the Fenton reaction to produce ROS. High concentrations of cellular ROS can cause serious lipid peroxidation in the cell membrane, thus leading to ferroptosis. Unexpectedly, PEI-HBP1 and MPN-HBP1 also decreased GPX4 protein levels and increased ACSL4 and TFRC protein levels but had no effect on their mRNA levels, indicating that HBP1 may regulate the expression of these proteins at the posttranscriptional level. To detect whether HBP1 promotes ferroptosis by other means than CDO1, we constructed HepG2 cell lines with both CDO1 knocked out and HBP1 overexpressed to detect cell viability. The results showed that HBP1 induced ferroptosis mainly through CDO1, but a small part of ferroptosis may also be induced by other pathways. Therefore, we speculate that HBP1 may also promote ferroptosis by regulating GPX4, ACSL4, and TFRC protein levels. The specific regulation mechanism needs further study.

Long-term anticancer experiments further indicated that MPN-HBP1 treatment not only suppresses tumor growth but also reduces systemic toxicity in tumor-bearing mice.

Consequently, nanoparticles that exert ferroptosis induction capability may bypass the drug tolerance that occurs with conventional anticancer agents and provide new clinical insight into tumor therapy where traditional therapeutic strategies have failed. Compared with most traditional drug delivery systems, this metal polyphenol coordination network (MPN) significantly improves the anticancer effect and enriches the therapeutic function of synthetic nanomedicine. This pioneering work revealed the mechanisms underlying ferroptosis processes and provides a new strategy for the development of therapeutic nanomedicine, which has great potential for the field of cancer treatment.

## Materials and methods

### Ethics statement

The study for mice is compliant with all relevant ethical regulations for animal experiments. All experiments and facilities were approved by the Committee for Ethics of Animal Experiments and were conducted in conformity to the Guidelines for Animal Experiments, Peking University Health Science Center (LA2020230). The study for human cancer samples complied with the ethical requirements of the Research Ethics Committee of Kunming Medical University (LL-2021-163-K). All participants gave oral informed consent.

### Reagents

Erastin, RSL-3, and necrostatin-1 (Ner-1) were purchased from Selleck Chemicals (Houston, USA). Tannic acid (TA), ferric chloride hexahydrate ($FeCl_3.6H_2O$), DMSO, and Ac-DEVD-CHO (Apo) were purchased from Macklin (Shanghai, China). EDTA and urea were purchased from Solarbio (Beijing, China). Fer-1, DFO, GSH, BSO, Liproxstatin-1, SF1670, and Chloroquine were purchased from MedChemExpress. Branched polyethyleneimine (PEI, Mw = 25,000) and 2′,7′-Dichlorofluorescin diacetate were purchased from Sigma-Aldrich. Antibodies against HA (MMS101P, Covance), FLAG (F1804, Sigma), UHRF1 (D6G8E, CST), HBP1 (11746-1-AP, Proteintech), CDO1 (12589–1 AP, Proteintech), PTEN (9188, CST), p-AKT (4060, CST), AKT (9272, CST), GPX4 (sc-166570, Santa cruz), ACSL4 (sc-365230, Santa cruz), TFRC (sc-32272, Santa cruz), β-actin (AC026, ABclonal). Rabbit anti-4-HNE antibody (Bioss). The following secondary antibodies were used: anti-mouse IgG antibody DyLight 800 (610-145-121, Rockland) and anti-rabbit IgG DyLight 800 (611-145-002, Rockland).

### Cell culture, transfection, lentivirus infection, and CRISPR/Cas9

HEK293T, HeLa, and HepG2 cells were cultured in the Dulbecco's Modified Eagle Medium (DMEM) supplemented with 10% fetal bovine serum (FBS), 100 U/mL penicillin, and 100 μg/mL streptomycin at 37°C under 5% $CO_2$ circumstance. Cell transfection were utilizing TurboFect transfection reagent (Thermo Scientific) referring to the manufacturer's instruction. The transfection efficiency was confirmed after 48 h posttransfection. The lentivirus plasmid pLL3.7-shHBP1 or pLL3.7-shUHRF1 were transfected to obtain the shHBP1 or shUHRF1 stable cell line. The primers' sequences used in this study were listed in the Supporting information (S1 Table). PLVX-IRES was inserted with HBP1, UHRF1, and CDO1 to acquire the stable expression in HepG2 and HeLa cells. Cells were selected for 1 week with puromycin and tested the positive efficiency.

For the design of sgRNA in ZhangFeng library, sgRNA targets CDO1 5′-CCATTTCCTT GATCCACAAGATT-3′, which was inserted in CRISPR vector Cas9−puro-PX459. HepG2 and HeLa cells were transfected with the expression plasmid and were selected with puromycin. Cells were cultured for 2 weeks and detected the effects of knockout expression.

## Western blotting

Cells were lysed in RIPA buffer (Thermo Scientific) including protease inhibitor cocktail (Sigma), and protein concentrations were measured utilizing the BCA protein assay kit (Pierce). A total of 25 to 50 μg of protein was separated by SDS-PAGE and transferred to nitro-cellulose membranes (Pall). Odyssey infrared imaging system (LI-COR Bioscience, Lincoln, NE) was used to acquire the infrared fluorescence image.

## Real-time PCR

Total RNA was extracted using the RNAsimple Total RNA kit (Tiangen). Quantitative RT-PCR and real-time PCR were performed utilizing ReverAid FirstStrand cDNA Synthesis kit (Thermo Scientific) and Maxima SYBR Green qPCR Master Mix (Thermo Scientific) according to the manufacturer's instructions. The mRNA expression was normalized by *GAPDH*. The primers' sequences used in this study were listed in the Supporting information (S1 Table).

## Reporter gene assay

HEK293T cells were plated on 24-well plates and transfected with plasmids for individual experiments about 36 to 48 h before harvesting. The luciferase activity was measured by the Dual-Luciferase Reporter Assay kit (Promega). Each group is provided with 3 parallel holes, and each assay was performed at least 3 times.

## Chromatin immunoprecipitation (ChIP)

ChIP assays were performed as described previously [10]. The ChIP primer sequences were listed in the Supporting information (S1 Table).

## In vivo ubiquitination assay

HEK293T cells transfected with HA-HBP1 were treated with or without Erastin for 24 h. We harvested and lysed cells by IP lysis buffer. The lysates were incubated with anti-HA antibody and protein ASepharose (GE Healthcare). Western blotting was performed with the indicated antibodies.

## Protein half-life assay

Each dish was added with cycloheximide treatment maintained for 0, 30, 60, 90, and 120 min separately at final concentration of 100 μg/ml. Then, we collected and lysed cells and analyzed the relevant protein expression by western blotting utilizing an anti-HBP1 antibody. Quantification of expression of HBP1 protein under different time points was determined utilizing Image J software and normalized to β-actin.

## Measurement of NADPH/NADP$^+$ ratio

The measurement of NADPH/NADP$^+$ ratio was determined using a NADP$^+$/NADPH Assay Kit with WST-8 (S0179, Beyotime, China) according the manufacturer's instructions. In brief, $1 \times 10^6$ cells cultured in DMEM were collected and lysed by 3 frozen–thaw cycles. Lysate sample was then separated into 2 portions. One portion was heated at 60°C to deplete NADP$^+$ (only NADPH left), while the other portion was left on ice as unheated sample (containing both NADP$^+$ and NADPH). NADP$^+$ could be reduced into NADPH in the Working buffer, and the NADPH formed further reduced WST-8 to formazan. The orange product (formazan)

was then measured at 450 nm spectrophotometrically. The NADPH/NADP$^+$ ratio was calculated using following formula: (intensity of heated sample) / (intensity of unheated sample − intensity of heated sample). The results were normalized by protein concentration of each sample.

## Transmission electron microscope assay

Treated cells were fixed with 1 mL general fixative (containing 2.5% glutaraldehyde in 0.1 M PBS) at 4°C and overnight. After washed with PBS for 3 times, the cells were further stained with 4% osmium tetroxide for 0.5 h at room temperature. After rinsed with distilled water, the cells were dehydrated at room temperature in a graded ethanol series of 30%, 50%, 70%, and 90% and followed by 3 rinses of 100% ethanol for 10 min each. After dehydration, cells were embedded in epoxy resin and the resin was stored at 55°C for 48 h to allow resin polymerization. The embedded samples were then sliced with a thickness of 50 to 70 nm. Finally, the cell sections were stained with 5% uranyl acetate for 15 min and 2% lead citrate for 15 min before TEM imaging.

## Intracellular ROS measurements

Cells were treated with different concentrations of test compound or drug. After 24 h, cells were incubated at a final concentration of 10 μM 2′,7′-DCFH-DA for 30 min at 37°C, after which they were washed, fixed with 4% paraformaldehyde for 15 min, and permeabilized (using 0.2% Triton X-100 in PBS) for 15 min under room temperature, dyed with DAPI, and immediately analyzed for fluorescence intensity under Leica Confocal Microscope (TCS SP8, Germany) with a ×63 objective lens.

If analyzed by flow cytometry, treated cells were washed 3 times with PBS, harvested, and suspended in PBS followed by flow cytometric analysis (Ex: 488 nm/Em: 510–555 nm).

If analyzed cells with BODIPY-C11 fluorescent dye, $1 \times 10^6$ cells were incubated with 5 μM BODIPY-C11 (Thermo Fisher Scientific) for 30 min at 37°C. Cells were washed 3 times with PBS, harvested, and suspended in serum-free medium followed by flow cytometric analysis (Ex: 488 nm/Em: 510–555 nm).

## Detection of mitochondrial ROS and mitochondrial membrane potential

Mitochondrial ROS was determined according the manufacturer's instructions. About $1 \times 10^4$ cells/well were plated onto 96-well plate (black with clear bottom). The cells were incubated with 5 μM MitoSOX Red Mitochondrial Superoxide Indicator (Invitrogen, USA) for 30 min at 37°C, 48 h later. Cells were then washed with PBS for 3 times. The fluorescence was read at 510/580 nm of Ex/Em using a fluorescence microplate reader.

JC-1 Staining Kit was used to observe visually ΔΨm of cells. About $1 \times 10^4$ cells/well were plated onto 96-well plate (black with clear bottom). Cells were stained with JC-1 kit (Beyotime, China) according to the manufacturer's protocol and observed with fluorescence microscope, 48 h later.

## Total antioxidant capacity assay

Total antioxidant capacity assay was performed with the Total Antioxidant Capacity Assay Kit with a Rapid ABTS method (S0121, Beyotime, China) according the manufacturer;s instructions.

## Cell viability assay

HeLa cells were seeded in the 96-well plate ($5 \times 10^3$ cells/well) for 24 h; MPN-HBP1 (with a HBP1 content of 2 µg/mL) or PEI-HBP1 (with a HBP1 content of 2 µg/mL) were added into each well. Immediately after the addition of MPN-HBP1 or PEI-HBP1, ferroptosis inhibitor ferrostatin-1 (10 µM at the final concentration), apoptosis inhibitor (Ac-DEVD-CHO, 59 µM), necroptosis inhibitor (Necrostatin-1, 20 µM), and autophagy inhibitor (Chloroquine, 10 µM) were also added into MPN-HBP1- or PEI-HBP1-treated cells. After coincubation for 24 h, the medium was replaced with 200 µL fresh medium and 15 µL MTT (5 mg/mL in PBS buffer) solution was added to each well. The medium was replaced with 200 µL DMSO, 4 h later. The absorbance at 490 nm was measured and the cell viability was calculated as (OD490 sample/ OD490 control) × 100%, where OD490 control was obtained in the absence of complexes and OD490 sample was obtained after coincubation with complexes. Concentrated ferrostatin-1, Chloroquine, Necrostatin-1, and Ac-DEVD-CHO were stocked in DMSO before used.

HepG2 cells were seeded in the 96-well plate ($5 \times 10^3$ cells/well) for 24 h; Erastin (10 µM) and GSH (10 µM) or Erastin (10 µM) and BSO (10 µM) were added into each well. After coincubation for 24 h, MTT was used to test cell activity.

## The measurement of intracellular $Fe^{2+}$ content

The cells were trypsinized and transferred to confocal well in a volume of 100 µl with a cell density of $1 \times 10^4$/ml. The cells were incubated with 1 µmol/L FerroOrange solution (Dojindo Molecular Technologies, Kumamoto, Japan) for 30 min and then observed under a confocal fluorescence microscope.

According to the manufacturer's instructions, use the Iron Assay kit (Abcam) to determine the intracellular ferrous iron content. Collected $5 \times 10^6$ cells to be tested were lysed with iron buffer, centrifuged at 12,000 rpm for 10 min to collect the supernatant, incubated with iron reducer reagent at 37°C for 30 min, and then incubated with iron probe at 37°C in dark for 1 h. The absorbance at 593 nm was measured with Microplate Reader.

## Detection of malondialdehyde (MDA)

Analysis of lipid peroxidation was assessed by quantification of MDA concentration in cell lysates using Lipid Peroxidation MDA Assay Kit (S0131, Beyotime) in accordance with the manufacturer's instructions.

## GSH and GSH/GSSH assay measurement

Briefly, $1 \times 10^6$ cell lysates were determined using a commercial GSH and GSSG Assay Kit (S0053, Beyotime, China) according the manufacturer's instructions.

## Methylation-specific PCR

Genomic DNA was isolated using the TIANamp Genomic DNA kit (Tiangen). Around 1 mg of DNA was bisulfate treated using EZ DNA methylation Gold (Zymo Research). Methylation-specific PCR (MSP) of CDO1 using primer sets were designed as follows: methylation-specific primers 5′-TTTTTGGGACGTCGGAGATAAC-3′, 5′-CGAAAAAACCCTACGAA-CACG-3′. Un-methylation-specific primers were as follows: CDO1, 5′-GATTTTTGG-GATGTTGGAGATAAT-3′, 5′-AAAACAAAAAAACCCTACAAACACA-3′. The PCR was carried out using the SYBR green master mix (Bio-Rad).

## MTT assay and EdU incorporation assay

Stable transfected cell sublines were constructed by lentivirus infection. Cells were seeded at a density of 1,000 cells per well into 96-well plates. After culturing for 1 to 7 days separately, 15 μl of MTT solution (5 mg/ml) was added to each well and incubated at 37°C lasting for 4 h. Then, we removed the medium and added 200 μl DMSO to each well to dissolve the formazan crystals. The samples were measured by the intensity of absorbance at 490 nm utilizing the microplate reader.

We used the EdU kit (Ribobio) according to manufacturer's instructions, and cells were photographed using the fluorescence microscopy (Leica). We counted at least 5 randomly chosen fields for statistical analysis.

## Tumorigenicity in nude mice

Stable transfected cell sublines were constructed by lentivirus infection. About $2 \times 10^6$ cells were suspended in 100 μl of PBS. Five-week-old male nude mice were subcutaneously injected with these cells. After injection for 4 weeks, the mice were killed by cervical dislocation, and the tumors were weighed and measured. All experiments and facilities were approved by the Committee for Ethics of Animal Experiments and were conducted in conformity to the Guidelines for Animal Experiments, Peking University Health Science Center (LA2020230).

## Preparation of PEI-HBP1 and MPN-HBP1

Herein, pLVX-IRES-puro-HBP1 plasmid was used for anticancer study. About 2 μg HBP1 plasmid was added to 3.97 μg PEI (25 kDa, 1 μg/μL) (at N/P ratio of 15) and diluted with 10 mM PBS to 100 μL. The mixture was vortexed for 10 s and incubated at 37°C for 30 min to form PEI-HBP1 complexes. The complexes were used immediately after preparation. Before adding to cells, the mixture was dilute to 1 mL with complete medium. For preparation of TA-based MPN-HBP1, 5 μL FeCl$_3$ (4.8 mM) was added to the as-prepared PEI-HBP1 complexes and vortexed for 5 s, and 5 μL TA (4.8 mM) was added subsequently and vortexed for 5 s, then 20 μL Tris buffer (pH = 8.0) was added and vortexed for 5 s to obtained MPN-HBP1 nanocomplexes. Before adding to cells, the mixture was dilute to 1 mL with complete medium. To keep parameters consistent, same PEI-HBP1 doses, concentrations, and incubation times for in vitro experiments were used to evaluate the efficiency of MPN-HBP1.

## Fluorescein-labeling of plasmids

pLVX-IRES-puro-HBP1 plasmid was labeled by fluorescein probe using the Label-IT fluorescein labeling kit (Mirus Bio, USA) according to the manufacturer's protocol. Briefly, plasmid (1 mg/mL) and Label-IT reagents (100 μL/mL) were mixed in the buffer and then incubated at 37°C in a water bath for 1 h. The labeled DNA was purified by G50 micro-spin purification columns and stored at −20°C (protected from light).

## Lysosomal escape detection

Fluorescein-labelled HBP1 was incubated with different polymers and added into cells 8 h, 4 h, 2 h, and 1 h before final treatment. Then, treated cells were washed with cold PBS twice and stained with lysotracker Red (Beyotime, Shanghai) for 1 h. Next, we used Antifade Mounting Medium with Hoechst 33342 (Beyotime, Shanghai) to stain the nuclei and observed by confocal laser scanning microscopy (TCS, SP8, Leica, Germany).

## In vivo anticancer therapy

All animal experiments were approved by the Committee for Ethics of Animal Experiments and were conducted in conformity to the Guidelines for Animal Experiments, Peking University Health Science Center. For in vivo experiments, HepG2 cells ($5 \times 10^6$ cells/mouse) were subcutaneously injected in male Balb/c nude mice (5 weeks) for tumor inhibition assays. After the tumor volume had reached to 100 mm$^3$, mice were randomized divided into different groups with 5 mice each. Then, mice were IV injected with 100 μL of PBS, PEI-HBP1, MPN-vector, Erastin, and MPN-HBP1 at first, fifth, and 10th days. On the 35th day, all animals were killed by cervical dislocation as the tumor reached the size limitation according to the Institutional Animal Care and Use Committee (IACUC) approved guidelines (LA2020230).

## Patient samples and immunohistochemistry staining

Tumor tissue samples with clinical information (obtained from the Department of Pathology, First Affiliated Hospital of Kunming Medical University with the informed consent and the approval from the Research Ethics Committee of Kunming Medical University) were analyzed, LL-2021-163-K. All participants gave oral informed consent.

The patient samples first deparaffinized and immersed for 10 min in PBS. The sections were heated in microwave oven for 5 min in Citrate Antigen Retrieval buffer for 3 times. After cooling to room temperature, endogenous peroxidase activity was blocked with 3% $H_2O_2$ and then incubated with 10% goat serum for blocking nonspecific staining for 30 min. The sections were subsequently incubated with primary antibodies overnight at 4˚C (HBP1 and UHRF1 were 1:100 diluted) and treated for 30 min with indicated second antibodies, respectively. The sections were then exposed for 3 min to DAB and rinsed off in deionized water to terminate DAB reaction. The evaluation of the IHC staining was performed by pathologist. Collectively, no staining (<5%) was scored as "−", weak staining (5% to 25%) was scored as "+", moderate staining (25% to 65%) was scored as "++", and strong staining (>65%) was scored as "+++".

## TCGA and computational data analysis

To analyze the correlations between HBP1 gene expression and UHRF1 gene expression, we used TCGA cancer RNA-seq data downloaded from the UCSC Xena browser (https://xenabrowser.net/). The correlation between HBP1 and UHRF1 in TCGA patients was analyzed by the Pearson correlation test.

Data for HCC patients were downloaded from LIHC data set (Cbioportal) of TCGA database. Patients from the database were divided into high HBP1 group and low HBP1 group according to the median HBP1 expression. We conduct GSEA on HBP1.

To analyze the correlations between HBP1 gene expression and drug response across cancer cell lines, we used the CTRP (http://portals.broadinstitute.org/ctrp.v2.1/). The AUC values of CTRP cell lines were downloaded from (https://github.com/remontoire-pac/ctrp-reference/tree/master/auc). The gene expression values of CTRP cell lines were downloaded from the Cancer Cell Line Encyclopedia (CCLE) data portal (https://portals.broadinstitute.org/ccle/data).

## Statistical analysis

Statistical analysis was performed using SPSS 22.0 software. Differences between 2 groups were calculated using a two-tailed Student $t$ test. One-way ANOVA was performed to assess differences among multiple groups. Pearson correlation analysis method was used to calculate the correlation between the expression of different genes. The data are reported as the

mean ± SD (standard error of mean) from at least 3 independent experiments, and $P < 0.05$ was considered significant. *, $p < 0.05$, **, $p < 0.01$, ***, $p < 0.001$.

## Supporting information

**S1 Fig. HBP1 regulates UHRF1 expression.** (A) HBP1 negatively correlated with UHRF1 mRNA levels in the TCGA cohort of LIHC, BRCA, LUAD, and LUSC. The linear relationship was determined by a Pearson correlation analysis. (B) GSEA plots of genes in high HBP1 expression group compared with low HBP1 expression group. High-rank gene sets are shown with the ES, normalized ES, and nominal $p$ valve. The underlying data for S1A and S1B Fig can be found in S1 Data. BRCA, Breast cancer; ES, enrichment score; GSEA, Gene Set Enrichment Analysis; HBP1, HMG box-containing protein 1; LIHC, liver hepatocellular carcinoma; LUAD, lung adenocarcinoma; LUSC, lung squamous cell carcinoma; TCGA, The Cancer Genome Atlas; UHRF1, ubiquitin-like with PHD and RING finger domains 1.
(PDF)

**S2 Fig. HBP1 reduces cellular antioxidant capacity and therefore sensitizes tumor cells to ferroptosis.** (A) NADPH/NADP$^+$ ratio and (B) GSH/GSSG ratio were measured in the indicated cells. (C) Intracellular ROS of the indicated cells were stained by DCFH-DA and determined by FCM. (D) Mitochondrial ROS were stained by MitoSOX Red and measured by fluorescence microplate reader. The fold changes of ROS levels relative to controls were shown. (E) The cells with depolarized mitochondria are represented as the cells that have lost ΔΨm. The proportions of the cell with depolarized mitochondria in the indicated cells are shown. For NAC treatment in (D) and (E), HepG2/HBP1 cells were pretreated with 100 nM NAC for 24 h before collection. (F) Total antioxidant capacity in the indicated cells was detected. (G) Cell death was determined in the indicated cell with treatment of the incremental doses of $H_2O_2$ for 24 h. (H) Calculation of z-scored Pearson correlation coefficients between small-molecule sensitivity data, expressed as AUCs, with basal gene-expression measurements, expressed as log2 robust-multi-array-average values. Green dot means the expression level of HBP1 gene was correlated with the sensitivity of RSL-3 (left panel). Correlation between HBP1 expression and RSL-3 sensitivity, based on the liver cancer cell lines ($n = 21$) from CTRP. Dose responses are normalized AUC values. The linear relationship was determined by a Pearson correlation analysis (right panel). (I) Viability of HeLa and HepG2 cells treated with different concentrations of Erastin or RSL-3. (J) HeLa cells were treated with Erastin or RSL3 in the absence or presence of ferrostatin-1 (5 μM or 10 μM) for 24 h, and then cell viability was measured. (K) Protein levels of HBP1 and UHRF1 in HeLa, HepG2, and Huh7 cells were treated with Erastin (10 μM) alone or in combination with DFO (50 μM). The underlying data for S2A–S2J Fig can be found in S1 Data. Error bars represent S.D. *, $p < 0.05$, **, $p < 0.01$, ***, $p < 0.001$. AUCs, areas under concentration–response curves; CTRP, Cancer Therapeutics Response Portal; DFO, Deferoxamine; FCM, flow cytometry; GSH, glutathione; HBP1, HMG box-containing protein 1; $H_2O_2$, hydrogen dioxide; NAC, N-acetyl-l-cysteine; ROS, reactive oxygen species.
(PDF)

**S3 Fig. Erastin extends the half-life of HBP1 protein.** HeLa cells were treated with Erastin for 24 h, and cells were incubated with the protein synthesis inhibitor CHX for 0, 30, 60, 90, or 120 min before collect. HBP1 and protein levels were detected by western blotting. Quantification of HBP1 protein levels was determined using Image J software normalized to β-actin. The underlying data for S3 Fig can be found in S1 Data. CHX, cycloheximide; HBP1, HMG box-containing protein 1.
(PDF)

**S4 Fig. HBP1 sensitizes tumor cells to ferroptosis by inhibiting UHRF1 expression in tumor cells.** (A) Cell viability was conducted with HepG2 cells stably transfected with vector, HBP1, HBP1+UHRF1 or vector, shHBP1, shHBP1+shUHRF1. (B, C) Indicated cells were treated with or without 10 μM Erastin/10 μM Sorafenib for 24 h. Cells were collected, and confocal was used to detect $Fe^{2+}$ levels and flow cytometry was used to detect BODIPY-C11 fluorescence signal for ROS. (D, E) Indicated cells were lysed and MDA content and GSH content were measured. Scale bar = 10 μm. The underlying data for S4A, S4B, S4D and S4E Fig can be found in S1 Data. Differences between 2 groups were calculated using a two-tailed Student $t$ test. One-way ANOVA was performed to assess differences among multiple groups. Error bars represent S.D. *, $p < 0.05$, **, $p < 0.01$, ***, $p < 0.001$. GSH, glutathione; HBP1, HMG box-containing protein 1; MDA, malondialdehyde; ROS, reactive oxygen species; UHRF1, ubiquitin-like with PHD and RING finger domains 1.
(PDF)

**S5 Fig. HBP1 does not affect the mRNA expression of ferroptosis-related genes except *CDO1*.** (A) Quantitative RT-PCR showing mRNA expression of ferroptosis-related genes in HBP1 overexpressed cells. (B) HBP1 did not binding to the endogenous *CDO1* promoter. ChIP assays were used to test the binding of exogenous HBP1 to endogenous *CDO1* gene. HEK293T cells were transfected with HA-HBP1. The region from position −456 to position −658 contains the predicted HBP1 affinity site and was analyzed by specific PCR. Anti-HA antibody was used in the indicated lanes. The underlying data for S5A Fig can be found in S1 Data. CDO1, cysteine dioxygenase 1; ChIP, chromatin immunoprecipitation; HBP1, HMG box-containing protein 1; RT-PCR, real-time PCR.
(PDF)

**S6 Fig. The HBP1-UHRF1-CDO1 axis inhibits tumor cell proliferation and tumorigenesis.** (A) CDO1 knockout promotes cell proliferation. HeLa, HepG2, and Huh7 cells with CDO1 knockout were analyzed by western blotting to detect the protein levels of CDO1, PTEN, p-AKT, and AKT. (B) CDO1 overexpression inhibits cell proliferation. HeLa, HepG2, and Huh7 cells with CDO1 overexpression were analyzed by western blotting to detect the protein levels of CDO1, PTEN, p-AKT, and AKT. (C) CDO1 can partially promote ferroptosis through PTEN-AKT signal pathway. HeLa, HepG2, and Huh7 cells with CDO1 overexpression treated with Erastin (10 μM) and SF1670 (10 μM) for 24 h. Cell viability was measured by MTT. (D, E) Indicated cells p-AKT/AKT protein ratio was determined using Image J software. (F) Indicated cells were treated with or without 10 μM Fer-1 for 24 h in the presence of Erastin (10 μM). Cell viability was measured using MTT. The underlying data for S6A–S6F Fig can be found in S1 Data. CDO1, cysteine dioxygenase 1; Fer-1, Ferrostatin-1; HBP1, HMG box-containing protein 1; p-AKT, phospho-AKT; UHRF1, ubiquitin-like with PHD and RING finger domains 1.
(PDF)

**S7 Fig. The nanoparticles MPN-HBP1 was designed to kill cancer cells via ferroptosis pathway.** (A) MPN-HBP1 dissociation by EDTA, NaCl, urea, and Tween 20. MPN-HBP1 was effectively dissociated by EDTA due to complexation competition. (B) HBP1, GPX4, ACSL4, and TFRC protein levels of PBS, PEI-HBP1, and MPN-HBP1 treated HeLa cells. (C, D) Tumor volume curves and tumor weight curves of experimental mice at the first 35 days. (E) Blood biochemistry indexes of experimental mice after IV injected with PBS, PEI-HBP1, Erastin, MPN-vector, and MPN-HBP1. The underlying data for S7C–S7E Fig can be found in S1 Data. ACSL4, acyl-CoA synthetase long chain family member 4; EDTA, ethylene diamine tetraacetic acid; GPX4, glutathione peroxidase 4; HBP1, HMG box-containing protein 1; IV,

intravenous injection; MPN, metal polyphenol network; PEI, polyethyleneimine; TFRC, transferrin receptor.
(PDF)

**S1 Table. Primers used in the experiments.**
(DOCX)

**S1 Data. Underlying data for Figs 1–7 and S1–S7.**
(XLSX)

**S1 Raw Images. Original scan images for Figs 1C, 1D, 2D, 2E, 3A–3G, 3J, 4A, 5C, 5D, 5E, 5H, 5K, 6E, 6F, 7E, S2K, S5B, S9A, S6A, S6B, and S7B.**
(PDF)

## Acknowledgments

We are grateful for the generous gift of pcDNA3.1-Flag-UHRF1 plasmids from Dr. Jiemin Wong (East China Normal University).

## Author Contributions

**Data curation:** Tongjia Zhang.

**Formal analysis:** Jiyin Wang.

**Investigation:** Ruixiang Yang.

**Methodology:** Yue Zhou, Shujie Wang.

**Software:** Tongjia Zhang, Yuning Cheng, Zhe Yang.

**Validation:** Hui Li.

**Visualization:** Wei Jiang.

**Writing – review & editing:** Xiaowei Zhang.

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
