## [Editor Report · Decision Letter 0]

28 Sep 2022

Dear Dr Zhang, 

Thank you for submitting your manuscript entitled "The transcription factor HBP1 activates ferroptosis in tumor cells by regulating the UHRF1-CDO1 axis" for consideration as a Research Article by PLOS Biology. Please accept my sincere apologies for the great delay in getting back to you as we consulted with an academic editor about your submission.

Your manuscript has now been evaluated by the PLOS Biology editorial staff, as well as by an academic editor with relevant expertise, and I am writing to let you know that we would like to send your submission out for external peer review.

Once your full submission is complete, your paper will undergo a series of checks in preparation for peer review. After your manuscript has passed the checks it will be sent out for review. To provide the metadata for your submission, please Login to Editorial Manager (https://www.editorialmanager.com/pbiology) within two working days, i.e. by Sep 30 2022 11:59PM.

Kind regards,

Richard

Richard Hodge, PhD

Associate Editor, PLOS Biology

rhodge@plos.org

PLOS

---

## [Decision Letter · Decision Letter 1]

1 Dec 2022

Dear Dr Zhang,

Thank you for your patience while your manuscript "The transcription factor HBP1 activates ferroptosis in tumor cells by regulating the UHRF1-CDO1 axis" was peer-reviewed at PLOS Biology. Please accept my apologies for the long delays that you have experienced during the peer review process. Your manuscript has been evaluated by the PLOS Biology editors, an Academic Editor with relevant expertise, and by two independent reviewers.

The reviews are attached below. As you will see, the reviewers find your manuscript interesting but suggest several experiments to strengthen the mechanistic insights into the direct role of CDO1 in the regulation of ferroptosis and tumour progression. In addition, Reviewer #2 raises concerns with the overall physiological context and relevance of the study, as well as the lack of controls for the knockdown experiments and effect sizes on the ferroptosis phenotypes. 

Based on the reviews, it is clear that a substantial amount of work would be required to meet the criteria for publication in PLOS Biology. However, given our and the reviewer interest in your study, we would be open to inviting a comprehensive revision of the study that thoroughly addresses all the reviewers' comments. Given the extent of revision that would be needed, we cannot make a decision about publication until we have seen the revised manuscript and your response to the reviewers' comments. Your revised manuscript would need to be seen by the reviewers again, but please note that we would not engage them unless their main concerns have been addressed.

We appreciate that these requests represent a great deal of extra work, and we are willing to relax our standard revision time to allow you 6 months to revise your study. Please email us (plosbiology@plos.org) if you have any questions or concerns, or envision needing a (short) extension.

**IMPORTANT - SUBMITTING YOUR REVISION**

*Resubmission Checklist*

*Published Peer Review*

*PLOS Data Policy*

*Blot and Gel Data Policy*

Sincerely,

Richard

Richard Hodge, PhD

Associate Editor, PLOS Biology

rhodge@plos.org

REVIEWS:

Reviewer #1: In this manuscript, Yang and colleagues address that HBP1-UHRF1-CDO1 axis promotes the progression of ferroptosis and inhibits the proliferation of tumor cells and tumorigenesis. Ferroptosis induction to tumor suppression has been implicated as a new perspective for further research on anti-cancer therapy and holds great potential. The authors demonstrate that the transcription factor HBP1 represses UHRF1 expression by binding to an affinity site in the UHRF1 promoter through constructing YP191/192AA and YT478/479AA. They find that down-regulating the level of UHFR1 decreases the methylation level of the CDO1 promoter region and promotes CDO1 expression. As a result, the authors conclude that an increased level of CDO1 inhibits tumor cell proliferation through the activation of PTEN/PI3K/AKT signaling pathway because the key protein levels changed in CDO1 KO cell lines. The authors use real-time PCR, chromatin immunoprecipitation, immunohistochemistry staining, and transmission electron microscopy, demonstrating that HBP1-UHRF1-CDO1 axis leads to the deficiency of GSH synthesis, elevated lipid peroxidation and ferroptosis in tumor cells and generate the MPN-HBP1 nanodrugs to enhance the effect of tumor therapy. Overall, this is an interesting study, since the finding of HBP1-UHRF-CDO1 pathway gives us more insights into ferroptosis regulation by CDO1. However, the exact role of CDO1 in the regulation of tumor progression remains elusive. 

Major points:

1) Fer1 treatment blocked HBP1 upregulation by Ferroptosis-inducing agents (FINs) as shown in Fig.3E, does iron chelator like DFO also has the similar effect?

2) FINs suppressed proteasome-mediated HBP1 degradation as shown in Fig3 F-H. The authors need also check protein stability of HBP1 in the presence of FINs. 

3) CDO1 is known to reduce GSH and promote ferroptosis, the authors need to prove GSH level is the key to mediate HBP1-regulated ferroptosis using GSH synthesis inhibitor like BSO or supplementing GSH to retore the endogenous GSH level. 

4) CDO1 also regulated PTEN/AKT signaling. Does this contribute to its regulation of ferroptosis, since AKT is known to regulate ferroptosis sensitivity through mTORC1 signaling. 

5) The authors need to prove the role of ferroptosis regulation by HBP1-UHRF-CDO1 in the tumor development via addition of ferroptosis inhibitor in vivo like Liprostatin-1. The current data couldn't demonstrate that HBP1-UHRF1-CDO1 axis suppresses tumor growth through promoting ferroptosis.

6) Does HBP1 regulate ferroptosis and tumor growth solely through CDO1?

7) In Fig.7F, how about the impacts of PEI-HBP1 and MPN-HBP1 on UHRF and CDO1 expression? It seems that they also decrease GPX4 and increase ACSL4 and TFRC, the result of which could also promote ferroptosis independent of CDO1 and GSH. How to explain their effects?

Reviewer #2: "The transcription factor HBP1 activates ferroptosis in tumor cells by regulating the UHRF1-CDO1 axis". The manuscript proposes a pro-ferroptosis role of HBP1 by affecting the expression of UHRF1, which, in turn affect the CDO1 promoter methylation and expression. In addition, they have showed the iron-containing nano-particles with HBP1 can reduce the xenografts by triggering ferroptosis. While the proposed pathways have some novelties, there are some concerns about the biological context and quality of the experimental data.

Main critiques:

1. The main data on the effects of ferroptosis are quite modest. Given the strong phenotypes of ferroptosis, the proposed data are not compelling to support the hypothetic pathways.

2. Most of the knockdown experiments were done with one single shRNA without verifying by independent shRNAs or sgRNAs. The authors do not seem to be concerned about the potential off-target effects of these tools reported to have the potential for significant off-target effects.

3. The physiological function of the HBP1 was not known. In what context would this regulatory axis be relevant ? No physiological or pathological contexts are provided.

4. Most of the experiments were done in Hela and HepG2. Should the authors wish to study the relevance of the HCC, maybe more HCC cells should be used.

---

## [Decision Letter · Decision Letter 2]

12 May 2023

Dear Dr Zhang,

Thank you for your patience while we considered your revised manuscript "The transcription factor HBP1 activates ferroptosis in tumor cells by regulating the UHRF1-CDO1 axis" for publication as a Research Article at PLOS Biology. This revised version of your manuscript has been evaluated by the PLOS Biology editors, the Academic Editor and the original reviewers.

Based on the reviews, I am pleased to say that we are likely to accept this manuscript for publication, provided you satisfactorily address the following data and other policy-related requests that I have provided below (A-H):

(A)We would like to suggest the following minor modification to the title:

“The transcription factor HBP1 promotes ferroptosis in tumor cells by regulating the UHRF1-CDO1 axis”

(B) In the human ethics statement in the Methods section, please provide the specific approval number issued by your ethics committee to conduct the study. In addition, please specify what type of informed consent you received from the participants (e.g. written or oral).

(C) In the animal ethics statement in the Methods section, please provide the specific approval number issued by your ethics committee to conduct the study. We would also be grateful if you could confirm whether or not your animal ethics committee is an Institutional Animal Care and Use Committee (IACUC) in both the ‘In vivo anti-cancer therapy’ and ‘Tumorigencity in nude mice’ sections. Finally, please provide the method of euthanasia used to sacrifice the mice for both animal experiments. 

(D) For figures containing Flow Cytometry data (Figure 4E, 7F, S2C, S4C), please provide the FCS files and a picture showing the successive plots and gates that were applied to the FCS files to generate the figure. We ask that you please deposit this data in the FlowRepository (https://flowrepository.org/) and provide the accession number/URL of the deposition in the Data Availability Statement in the online submission form.

(E) Please also ensure that each of the relevant figure legends in your manuscript include information on *WHERE THE UNDERLYING DATA CAN BE FOUND*, and ensure your supplemental data file/s has a legend.

(F) Thank you for already providing the original and uncropped images of the gel/blot data in the following figures: 

Figure 3A, S1B-E, S2G, S2I, S5A-B, S6A-L, S7D-E

However, we note that some of the images are not completely uncropped. If the blots were cut during the course of the blotting process, then this is fine, but if you have the uncropped images then we ask that you please include/replace them in the S1_Raw Images file. In addition, we note that the blot for Fig S7B in the S1_Raw Images may be mislabeled (labelled as S7E, should be S7B?) and there does not appear to be a blot presented in Fig S5B in the manuscript?

(G) Please ensure that your Data Statement in the submission system accurately describes where your data can be found and is in final format, as it will be published as written there.

(H) Please note that per journal policy, the model system/species studied should be clearly stated in the abstract of your manuscript. 

We expect to receive your revised manuscript within two weeks. 

*Published Peer Review History*

*Press*

Sincerely,

Richard

Richard Hodge, PhD

Associate Editor, PLOS Biology

rhodge@plos.org

Reviewer remarks:

Reviewer #1 (Yilei Zhang, signs review): The review has no further questions regarding the current manuscript. 

Reviewer #2: the authors have responded to my concerns.

---

## [Editor Report · Decision Letter 3]

31 May 2023

Dear Dr Zhang,

On behalf of my colleagues and the Academic Editor, Mathieu Bertrand, I am pleased to say that we can accept your manuscript for publication, provided you address any remaining formatting and reporting issues. These will be detailed in an email you should receive within 2-3 business days from our colleagues in the journal operations team; no action is required from you until then. Please note that we will not be able to formally accept your manuscript and schedule it for publication until you have completed any requested changes.

PRESS

Best wishes, 

Richard

Richard Hodge, PhD

Associate Editor, PLOS Biology

rhodge@plos.org

PLOS
